# Defective apoptotic cell contractility provokes sterile inflammation, leading to liver damage and tumour suppression

**Linda Julian[1,2†‡], Gregory Naylor[1,2†§], Grant R Wickman[1,2†#], Nicola Rath[1¶], Giovanni Castino[3], David Stevenson[1], Sheila Bryson[1], June Munro[1], Lynn McGarry[1], Margaret Mullin[4], Alistair Rice[5], Armandodel Del Río Hernández[5], Michael F Olson[3]\***

[1]Cancer Research United Kingdom Beatson Institute, Garscube Estate, Glasgow, United Kingdom; [2]Institute of Cancer Sciences, University of Glasgow, Glasgow, United Kingdom; [3]Department of Chemistry and Biology, Ryerson University, Toronto, Canada; [4]Electron Microscopy Facility, School of Life Sciences, University of Glasgow, Glasgow, United Kingdom; [5]Cellular and Molecular Biomechanics Laboratory, Department of Bioengineering, Imperial College London, London, United Kingdom

**\*For correspondence:** michael.olson@ryerson.ca

[†]These authors contributed equally to this work

**Present address:** [‡]Cancer Research UK Cambridge Institute, Robinson Way, Cambridge, United States; [§]Beatson West of Scotland Cancer Centre, NHS Greater Glasgow & Clyde, Glasgow, United Kingdom; [#]Zymeworks, Vancouver, Canada; [¶]Oncology R&D, AstraZeneca, Cambridge, United Kingdom

**Competing interests:** The authors declare that no competing interests exist.

**Abstract** Apoptosis is characterized by profound morphological changes, but their physiological purpose is unknown. To characterize the role of apoptotic cell contraction, ROCK1 was rendered caspase non-cleavable (ROCK1nc) by mutating aspartate 1113, which revealed that ROCK1 cleavage was necessary for forceful contraction and membrane blebbing. When homozygous ROCK1nc mice were treated with the liver-selective apoptotic stimulus of diethylnitrosamine, ROCK1nc mice had more profound liver damage with greater neutrophil infiltration than wild-type mice. Inhibition of the damage-associated molecular pattern protein HMGB1 or signalling by its cognate receptor TLR4 lowered neutrophil infiltration and reduced liver damage. ROCK1nc mice also developed fewer diethylnitrosamine-induced hepatocellular carcinoma (HCC) tumours, while HMGB1 inhibition increased HCC tumour numbers. Thus, ROCK1 activation and consequent cell contraction are required to limit sterile inflammation and damage amplification following tissue-scale cell death. Additionally, these findings reveal a previously unappreciated role for acute sterile inflammation as an efficient tumour-suppressive mechanism.

## Introduction

Apoptotic cell death and the efficient clearance of cell corpses are essential for the normal development, growth and maintenance of every tissue in the human body (*Elliott and Ravichandran, 2010*; *Boada-Romero et al., 2020*). The biochemical events leading from the sensation of intrinsic or extrinsic apoptotic stimuli to the execution of cell death have been thoroughly characterized (*Elmore, 2007*; *Nagata, 2018*). And yet, the purpose of the apoptotic morphological events of cell contraction, membrane blebbing and apoptotic body formation, which are the most obvious, earliest observed and defining hallmarks of apoptotic cell death, remains elusive. It has long been assumed that these apoptotic morphological changes are required for efficient clearance of apoptotic cell debris by phagocytic cells (*Orlando et al., 2006*; *Wickman et al., 2012*), and that they limit the release of cellular contents that could provoke acute inflammatory responses and the development of autoimmunity over time (*Nagata et al., 2010*). Consistent with this concept, defects in the recognition and clearance of apoptotic cells (*Botto et al., 1998*; *Hanayama et al., 2004*) result in autoimmune diseases in genetically modified mouse models. However, whether the apoptotic

morphological responses are similarly essential for tissue homeostasis and organismal health have not been conclusively established.

Apoptotic cell death results from the activation of the cysteine-dependent aspartate-directed proteases (caspases) that cleave target proteins at specific recognition sequences. We and others previously discovered that during apoptosis caspase cleavage of the ROCK1 protein kinase, at a highly conserved DETD sequence (amino acids 1110–1113) not found in the close homologue ROCK2, removes a C-terminal autoinhibitory domain to produce a constitutively active fragment with increased phosphorylating activity towards substrates including regulatory myosin light chains (MLCs) (*Coleman et al., 2001*; *Sebbagh et al., 2001*). MLC phosphorylation enhances actin-myosin contraction (*Torgerson and McNiven, 1998*), and both intact actin filaments and MLC phosphorylation are required for apoptotic membrane blebbing (*Mills et al., 1998*; *Coleman and Olson, 2002*). Importantly, pharmacological inhibition revealed that ROCK contributed to MLC phosphorylation, cell contraction, membrane blebbing and nuclear fragmentation in apoptotic cells (*Coleman et al., 2001*; *Croft et al., 2005*; *Wickman et al., 2013*).

To establish the physiological importance of apoptotic ROCK1 activation by caspase cleavage, and the associated morphological events (e.g. contraction, blebbing), a genetically modified mouse was established in which a single amino acid in the ROCK1 caspase cleavage site was mutated (D1113A) to render the protein caspase-resistant. Analysis of mouse embryo fibroblasts (MEFs) expressing wild-type ROCK1 (ROCK1wt) or non-cleavable ROCK1 (ROCK1nc) revealed that caspase-mediated ROCK1 activation was essential for increased MLC phosphorylation and the generation of contractile force during apoptosis. While apoptotic ROCK1wt MEFs underwent typical contraction and membrane blebbing, ROCK1nc MEF death morphologically resembled necrosis, which was associated with greater release of cellular contents during late apoptosis. Surprisingly, ROCK1nc mice manifested no overt indications of autoimmunity, including glomerulonephritis, haemolytic anaemia, cytopenia, splenomegaly or splenic hemosiderin deposition, indicating that the altered morphological changes during the basal apoptotic turnover of cells in adult tissues were insufficient to provoke phenotypic differences. However, following tissue-scale chemical induction of hepatocyte apoptosis, ROCK1nc mice had significantly more liver damage and greater levels of neutrophil infiltration. Inhibition of the damage-associated molecular pattern (DAMP) protein high-mobility group B1 (HMGB1) or blocking signalling by its cognate pattern recognition receptor (PRR) toll-like receptor 4 (TLR4) significantly reduced neutrophil recruitment and liver damage in ROCK1nc mice, indicating that the ROCK1nc mutation led to greater sterile inflammation that amplified the initial hepatotoxin-induced injury. Interestingly, ROCK1nc mice developed fewer genotoxin-induced hepatocellular carcinoma (HCC) tumours than ROCK1wt mice, while HMGB1 inhibition concomitant with genotoxin administration increased tumour numbers. Although injury amplification may exacerbate the toxic effects of ingested xenobiotics, these findings reveal long-term benefits of acute sterile inflammation as an effective mechanism to suppress liver cancer. This study is the first to demonstrate that there is a critical role for ROCK1 cleavage and associated morphological events in limiting the deleterious effects of apoptotic cell death on tissue health.

## Results

### ROCK1 cleavage is essential for increased MLC phosphorylation during apoptosis

Although it had previously been shown that a ROCK1 fragment equivalent to the caspase-cleaved form was more active than full-length protein (*Coleman et al., 2001*), and that the increased MLC phosphorylation in apoptotic cells could be blocked by ROCK or caspase inhibitors (*Sebbagh et al., 2001*), direct ROCK1 activation solely by caspase cleavage had not been formally demonstrated. A recent study reported that ROCK2-specific activity was not affected by deletion of the C-terminal region (*Truebestein et al., 2015*), raising the possibility that in apoptotic cells ROCK1 cleavage plus an additional post-translational modification might be required for ROCK1 activation. While immunoprecipitated Myc-tagged kinase-dead (KD) ROCK1 K105M did not phosphorylate recombinant MLC in vitro, the ability of wild-type ROCK1 (ROCK1wt) to phosphorylate MLC was significantly increased approximately fivefold following in vitro cleavage by recombinant caspase 3 (*Figure 1A, Figure 1—figure supplement 1A*). Although basal activity of caspase-resistant non-cleavable

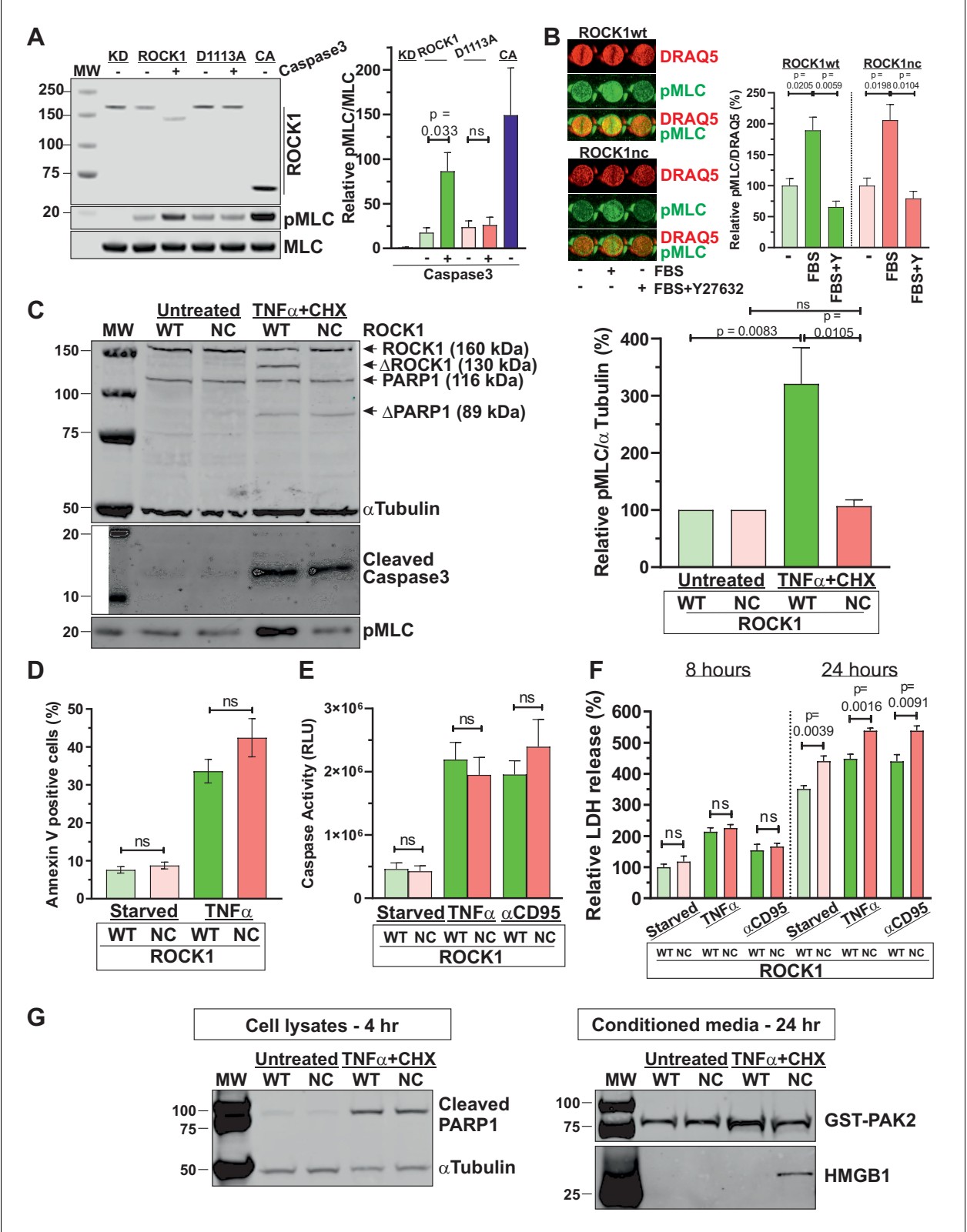

**Figure 1.** ROCK1 cleavage is essential for increased myosin light chain (MLC) phosphorylation during apoptosis. (**A**) Representative western blot showing Myc-tagged ROCK1 immunoprecipitated from transfected HEK293T cells assayed in vitro for MLC phosphorylation (pMLC) without (-) or with (+) cleavage by recombinant active caspase 3. KD: kinase-dead (K105A; *Ishizaki et al., 1997*); CA: catalytically active (ROCK1Δ2; *Ishizaki et al., 1997*). Means ± SEM pMLC/MLC relative to KD levels, n = 3 independent replicates. Pairwise Student's t-test between indicated conditions; ns: not significant.

*Figure 1 continued on next page*

Figure 1 continued

(B) Homozygous ROCK1wt (green) or ROCK1nc (red) mouse embryo fibroblasts (MEFs) were serum-starved overnight, then left untreated or stimulated for 5 min with medium containing 10% fetal bovine serum (FBS) without or with 10 μM Y27632 as indicated. Fixed cells were stained for pMLC and DNA using DRAQ5, and levels quantified by infrared quantitative imaging. Means ± SEM. pMLC/DRAQ5 levels relative to starved condition for each genotype, n = 3 independent replicates. Pairwise Student's t-test between indicated conditions. (C) Homozygous ROCK1wt (WT) or ROCK1nc (NC) MEFs were serum-starved overnight, then left untreated or treated with tumour necrosis factor α (TNFα) (50 ng/mL) plus cycloheximide (CHX; 10 μg/mL) for 4 hr. Representative blot shows quantitative western blot of ROCK1, cleaved ROCK1 (ΔROCK1), PARP1, cleaved PARP1 (ΔPARP1) α-tubulin, cleaved caspase 3 and pMLC as indicated, using corresponding infrared fluorophore-conjugated secondary antibodies. Means ± SEM. pMLC/α-tubulin levels relative to starved condition for each genotype, n = 5 independent replicates. Pairwise Student's t-test between indicated condition; ns: not significant. (D) Homozygous WT or NC MEFs were serum-starved overnight, then left untreated or treated with TNFα plus CHX for 4 hr to induce apoptosis. Phosphatidylserine externalization was measured using FITC-labelled Annexin V flow cytometry. Means ± SEM percentage FITC-labelled Annexin V-positive cells, n = 3 independent replicates. Pairwise Student's t-test within conditions between genotypes; ns: not significant.
(E) Homozygous WT or NC MEFs were serum-starved overnight, then left untreated or treated with TNFα plus CHX or anti-CD95 antibody (2 μg/mL) plus CHX (10 μg/mL) to induce apoptosis. Caspase activity was determined after 4 hr by Caspase-Glo 3/7 assays. Means ± SEM relative luminescence units (RLUs), n = 4 independent replicates. Pairwise Student's t-test within conditions between genotypes; ns: not significant. (F) Homozygous WT or NC MEFs were serum-starved overnight, then left untreated or treated with TNFα plus CHX or anti-CD95 antibody plus CHX to induce apoptosis. Lactate dehydrogenase (LDH) activity was measured after 8 or 24 hr by colorimetric assay. Means ± SEM percentage LDH activity in media normalized to cell density determined using crystal violet, then relative values calculated using the 8 hr starved WT condition as reference, n = 4 independent replicates. Pairwise Student's t-test between indicated conditions; ns: not significant. (G) Homozygous ROCK1wt or ROCK1nc MEFs were serum-starved overnight, then left untreated or treated with TNFα plus CHX for 4 or 24 hr. Left panel: representative western blot shows cleaved PARP1 and α-tubulin in cell lysates 4 hr after TNFα plus CHX treatment. Right panel: representative western blot shows recombinant GST-PAK2 used as a control for the efficiency of protein recovery with StrataClean resin, and HMGB1 in conditioned media 24 hr after TNFα plus CHX treatment.
The online version of this article includes the following figure supplement(s) for figure 1:

Figure supplement 1. Uncropped western blots and in-cell westerns.
Figure supplement 2. Homologous recombination targeting strategy to generate ROCK1 non-cleavable allele.
Figure supplement 3. Example fluorescence-activated cell sorting (FACS) determination of ROCK1wt and ROCK1nc mouse embryo fibroblast (MEF) apoptosis.

---

ROCK1 D1113A (ROCK1nc) was equivalent to wild-type ROCK1, it was neither proteolyzed nor activated by recombinant caspase 3. Consistent with previous findings (*Ishizaki et al., 1997*), a constitutively active (CA) ROCK1 fragment (ROCK1Δ2; amino acids 1–727) efficiently phosphorylated MLC. Thus, caspase-mediated removal of the autoinhibitory C-terminus is sufficient for ROCK1 activation.

To identify the physiological purpose of ROCK1 cleavage, and by extension of apoptotic cell contraction and membrane blebbing, genetically modified mice with the ROCK1 D1113A mutation were established (*Figure 1—figure supplement 2A–C*). Heterozygote ROCK1wt/nc breeding pairs (in C57Bl/6J backgrounds) produced offspring at expected Mendelian ratios (*Figure 1—figure supplement 2D*). Homozygous ROCK1nc/nc mice were apparently healthy and females were able to undergo multiple rounds of productive reproduction. Stimulation of serum-starved homozygous ROCK1wt or ROCK1nc MEFs for 5 min with media containing 10% fetal bovine serum (FBS) resulted in significant and comparable approximately twofold increases in phosphorylated MLC (pMLC) in a Y27632 ROCK inhibitor-sensitive manner (*Figure 1B, Figure 1—figure supplement 1B*), indicating that the ROCK1 D1113A mutant behaved similarly to wild-type ROCK1 in response to a physiological stimulus. Similarly, there were equivalent levels of basal MLC phosphorylation in serum-starved ROCK1wt and ROCK1nc MEFs (*Figure 1C, Figure 1—figure supplement 1C*). However, the induction of apoptosis by treatment with tumour necrosis factor α (TNFα) plus cycloheximide (CHX) for 4 hr resulted in ROCK1 cleavage and a significant approximately threefold increase in MLC phosphorylation in ROCK1wt but not ROCK1nc MEFs, despite comparable caspase 3 activation and PARP1 cleavage (*Figure 1C, Figure 1—figure supplement 1C*). These results demonstrate that ROCK1 cleavage is the primary driver of increased MLC phosphorylation in apoptotic cells.

ROCK1 cleavage, or MLC phosphorylation by association, were not required for biochemical processes that mediate apoptosis since the caspase-regulated process of phosphatidylserine (PS) externalization (*Segawa et al., 2014*) was comparably increased more than threefold in response to TNFα plus CHX treatment for 4 hr in ROCK1wt and ROCK1nc MEFs (*Figure 1D, Figure 1—figure supplement 3*). Similarly, caspase activation in response to TNFα plus CHX or antibody-mediated CD95 (Fas) activation plus CHX was equivalent after 4 hr in ROCK1wt and ROCK1nc MEFs (*Figure 1E*). These results are consistent with our previous observations that these apoptotic

processes were not affected in magnitude or kinetics by the ROCK selective inhibitor Y27632 (*Coleman et al., 2001*).

While necrosis results in the uncontrolled release of cellular contents that may act as DAMPs or alarmins to trigger inflammatory responses, cells in early apoptosis maintain their plasma membrane integrity and retain DAMPs while displaying 'eat-me' signals such as PS to facilitate recognition and internalization by phagocytes (*Wickman et al., 2012*; *Poon et al., 2010*). Over time, uncleared apoptotic cells may undergo secondary necrosis, leading to DAMP release. The cytoplasmic enzyme lactate dehydrogenase (LDH) activity in conditioned media can be assayed as a proxy for cell death, but more accurately reflects loss of membrane integrity and cytoplasmic protein release. Eight hours after serum starvation, TNFα plus CHX or anti-CD95 antibody plus CHX treatment, there were no significant differences in LDH release from ROCK1nc relative to ROCK1wt MEFs for each condition (*Figure 1F*). However, there was significantly more LDH released from ROCK1nc relative to ROCK1wt MEFs in each condition after 24 hr (*Figure 1F*). Furthermore, 4 hr after treatment of ROCK1wt and ROCK1nc MEFs with TNFα plus CHX resulted in comparable cleavage of the well-characterized caspase substrate PARP1 (*Soldani and Scovassi, 2002*; *Figure 1G, Figure 1—figure supplement 1D*). Using StrataClean resin to concentrate proteins released from apoptotic cells as we previously described (*Wickman et al., 2013*), equivalent amounts of recombinant GST-PAK2 that had been added to equal volumes of conditioned media as a control for protein recovery were detected (*Figure 1G, Figure 1—figure supplement 1D*). In contrast, recovery of the DAMP high-mobility group protein 1 (HMGB1) (*Scaffidi et al., 2002*) was markedly greater from apoptotic ROCK1nc MEFs relative to ROCK1wt MEFs (*Figure 1G, Figure 1—figure supplement 1D*). These observations indicate that the inability of caspases to activate ROCK1 and induce MLC phosphorylation results in morphological differences favouring secondary necrosis during late apoptosis.

## ROCK1 cleavage is required for apoptotic cell contraction

Given the central role of myosin in controlling cell morphology, the absence of increased MLC phosphorylation in apoptotic ROCK1nc MEFs could affect the rapid and forceful cell contraction that is a defining hallmark of apoptosis. Proliferation of ROCK1wt and ROCK1nc MEFs was comparable over 4 days (*Figure 2—figure supplement 1A*), and high content image analysis revealed MEFs from each genotype to be comparable for 10 nuclear and cellular morphological features (*Figure 2—figure supplement 1B–L*), consistent with the ROCK1nc mutation not impacting signalling or morphology regulation in healthy cells. Differential interference contrast (DIC) microscopy was used to acquire images at 3 min intervals after treatment of ROCK1wt or ROCK1nc MEFs with TNFα plus CHX to induce apoptosis, while scanning electron microscopy enabled detailed imaging of features on the surfaces of apoptotic cells. While ROCK1wt MEFs underwent rapid and forceful contraction that gave rise to characteristic apoptotic cells studded with membrane blebs (*Figure 2A*, upper panels; *Video 1*), ROCK1nc MEFs did not forcefully contract but appeared to collapse, leaving less orderly cell corpses (*Figure 2A*, lower panels; *Video 2*). To measure changes in contractile force over time, MEFs were plated on fibronectin-coated polydimethylsiloxane (PDMS) elastic micropillars (*Figure 2B*), then images were acquired every 2 s over a 10 min period during which cells observably contracted. The maximum force applied to each pillar (*Figure 2C*) and the mean maximum force per cell were calculated from measurements of the maximum displacement of each pillar covered by a cell over 10 min and the spring constant of the pillars (*Rice et al., 2016*). The maximum forces applied to each pillar can be displayed as a heat map (*Figure 2C*) or as a force map representing the direction and magnitude of the maximum forces (*Figure 2D*). As a control, mean maximum forces applied to pillars by non-apoptotic ROCK1wt and ROCK1nc MEFs were measured at random 10 min intervals (*Figure 2E*). In contrast to the significant forces generated during ROCK1wt apoptotic cell contraction, mean maximum forces applied by ROCK1nc cells during apoptotic cell death were not significantly different from basal forces generated by non-apoptotic ROCK1nc or ROCK1wt cells (*Figure 2D, E*). These results demonstrate that caspase cleavage and activation of ROCK1 is essential for the forceful contraction and membrane blebbing of apoptotic cells.

## Absence of autoimmunity in ROCK1nc mice

After backcrossing for at least 10 generations into the C67Bl/6J background, homozygous ROCK1wt and ROCK1nc mice were examined histologically to determine if the absence of ROCK1 activation

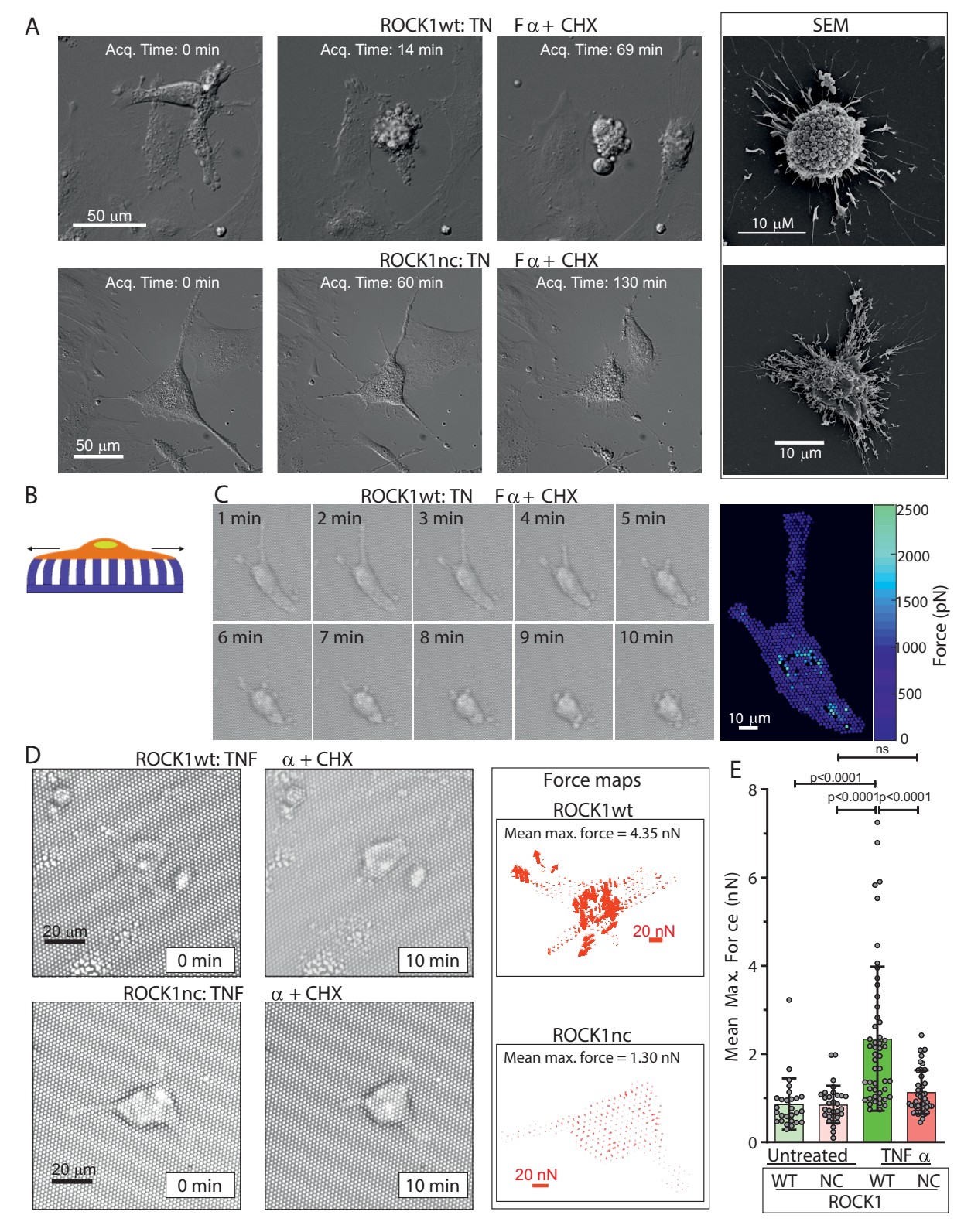

**Figure 2.** Apoptotic ROCK1nc mouse embryo fibroblasts (MEFs) do not undergo forceful contraction. (**A**) Homozygous ROCK1wt (upper panels) or ROCK1nc (lower panels) MEFs were serum-starved overnight, then treated with tumour necrosis factor α (TNFα) (50 ng/mL) plus cycloheximide (CHX) (10 μg/mL) to induce apoptosis. Differential interference contrast time-lapse microscopy images were acquired at 3 min intervals after the addition of TNFα plus CHX. Scale bar = 50 μm. Right panels: scanning electron microscope images. Scale bar = 10 μm. (**B**) Schematic diagram of a cell on elastic

*Figure 2 continued on next page*

*Figure 2 continued*

micropillars. (**C**) Homozygous ROCK1wt MEFs plated on elastic micropillars were serum-starved overnight, then treated with TNFα plus CHX to induce apoptosis. Brightfield time-lapse microscopy images were acquired at 2 s intervals after the commencement of cell contraction over 10 min. Right panel: force heat map indicating the magnitude of the force applied to each individual pillar. Scale bar = 10 µm. (**D**) Homozygous ROCK1wt (upper panels) or ROCK1nc (lower panels) MEFs were serum-starved overnight, then treated with TNFα plus CHX to induce apoptosis. Brightfield time-lapse microscopy images were acquired at 2 s intervals after the commencement of cell contraction over 10 min. Right panels: force vector map with direction and magnitude of the forces applied to each individual pillar. Scale bar = 20 nN. Mean maximum force is the average of the maximum forces applied to each pillar over 10 min. (**E**) Mean maximum force for untreated or TNFα plus CHX-treated homozygous ROCK1wt (green) or ROCK1nc (red) MEFs. Means ± SD n=27–50 individual cells. One-way ANOVA with *post-hoc* Tukey's multiple comparison test; ns: not significant.

The online version of this article includes the following figure supplement(s) for figure 2:

**Figure supplement 1.** Characterization of ROCK1wt and ROCK1nc mouse embryo fibroblasts (MEFs).

by caspase cleavage and consequent reduction in apoptotic cell contractile force generation pre-disposed mice to manifest characteristics of autoimmunity. Glomerulonephritis is a chronic kidney condition frequently observed in autoimmune diseases such as systemic lupus erythromatosus (*Anders et al., 2020*) and which has been linked with defective clearance of apoptotic cells (*Botto et al., 1998*). Periodic acid-Schiff (PAS) staining of kidney sections to highlight glomerular capillary basement membranes did not reveal observable differences that could have resulted from deposition of antibodies or immune complexes between ROCK1wt or ROCK1nc mice (*Figure 3—figure supplement 1A*). Nor were there apparent differences between the genotypes in glomerular hypercellularity in haematoxylin and eosin (H&E) stained kidney sections or detectable proteinuria (data not shown). Autoimmune haemolytic anaemia can lead to increased deposition by splenic macrophages of iron recovered from haemoglobin in damaged red blood cells in the form of hemosiderin (*Suttie, 2006*). Perls Prussian blue staining of spleen sections from ROCK1wt and ROCK1nc mice revealed that there were comparable levels of hemosiderin (*Figure 3—figure supplement 1B*). When mice were aged for 1 year, haematological analysis revealed no apparent autoimmune haemolytic anaemia or cytopenia as there were no significant differences between ROCK1wt and ROCK1nc mice at 10 weeks or 1 year for red blood cells (*Figure 3—figure supplement 2A*), white blood cells (*Figure 3—figure supplement 2B*), neutrophils (*Figure 3—figure supplement 2C*), lymphocytes (*Figure 3—figure supplement 2D*), monocytes (*Figure 3—figure supplement 2E*), eosinophils (*Figure 3—figure supplement 2F*) or platelets (*Figure 3—figure supplement 2G*). Furthermore, there was no indication of splenomegaly after 1 year, with comparable spleen to body mass ratios for ROCK1wt and ROCK1nc mice (*Figure 3—figure supplement 2H*). Similarly, liver to body mass ratios were equivalent between the genotypes (*Figure 3—figure supplement 2I*). Taken together, there were no overt indications of autoimmunity in unchallenged ROCK1nc mice.

## ROCK1 cleavage limits chemically induced liver damage

Given the absence of phenotype in unchallenged mice, ROCK1wt and ROCK1nc mice were examined in a model of chemically induced liver damage. The liver was selected as the target organ

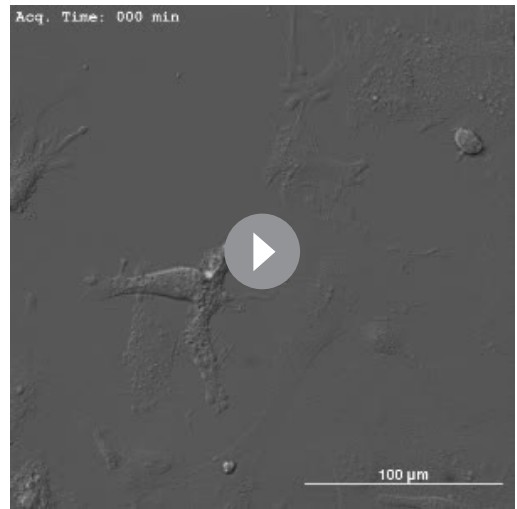

**Video 1.** Time lapse of apoptotic ROCKwt mouse embryo fibroblasts (MEFs). ROCK1wt MEFs were plated on 6-well glass-bottom dishes at a density of $2 \times 10^6$ cells per well. Cells were serum-starved overnight prior to induction of apoptosis with tumour necrosis factor α (TNFα) (50 ng/mL) and cycloheximide (10 µg/mL) diluted in serum-free starvation medium. Differential interference contrast (DIC) time-lapse microscopy images were acquired with a 20× DIC objective using a Nikon TE 2000 microscope with a heated stage and 5% $CO_2$ gas line. Immediately after induction of apoptosis the dishes were transferred to the microscope and time-lapse images were taken each minute.
https://elifesciences.org/articles/61983#video1

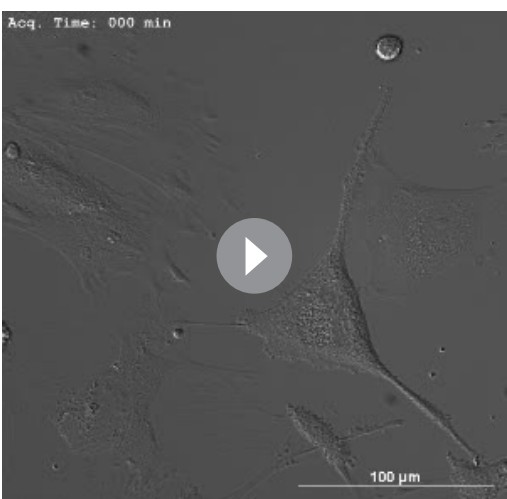

**Video 2.** Time lapse of apoptotic ROCKnc mouse embryo fibroblasts (MEFs). ROCK1nc MEFs were plated on 6-well glass-bottom dishes at a density of $2 \times 10^6$ cells per well. Cells were serum-starved overnight prior to induction of apoptosis with tumour necrosis factor $\alpha$ (TNF$\alpha$) (50 ng/mL) and cycloheximide (10 μg/mL) diluted in serum-free starvation medium. Differential interference contrast (DIC) time-lapse microscopy images were acquired with a 20× DIC objective using Nikon Eclipse TI microscope with a heated stage and 5% $CO_2$ gas line. Immediately after induction of apoptosis the dishes were transferred to the microscope and time-lapse images were taken each minute.

https://elifesciences.org/articles/61983#video2

because of its function as the centre for xenobiotic metabolism, which may result in apoptosis if exposure exceeds detoxification capacity (*Gu and Manautou, 2012*). To protect against organ failure, the liver has a remarkable capacity to clear apoptotic debris and to re-generate, recovering fully after removal/damage of up to 75% of the original tissue (*Gilgenkrantz and Collin de l'Hortet, 2018*). The liver-selective hepatotoxin diethylnitrosamine (DEN) (*Tolba et al., 2015*) undergoes CYP2E1-mediated oxidation (*Kang et al., 2007*) to become a genotoxin (*Bakiri and Wagner, 2013*), leading to hepatocyte apoptosis and tissue damage that can progress to cirrhosis and HCC (*Tolba et al., 2015*). DEN-induced liver damage and HCC occur with high penetrance in male mice, with female mice being relatively insensitive (*Tolba et al., 2015*), thus, all DEN studies were performed using 10-week-old male mice. After intraperitoneal (IP) injection with 100 mg/kg DEN, sera and livers were collected at 24 hr intervals for up to 4 days for analysis. The high efficiency of efferocytosis often makes it difficult to detect uncleared apoptotic cells (*Doran et al., 2020*). Nevertheless, apoptotic cells were detected in ROCK1wt and ROCK1nc livers 24 hr after DEN administration by terminal deoxynucleotidyl transferase dUTP nick end labelling (TUNEL) staining (*Figure 3A*). Although there was a reduction in TUNEL-positive cells in ROCK1wt livers after 72 hr, the number of TUNEL-positive cells increased in ROCK1nc livers relative to 48 hr and was significantly higher than in ROCK1wt livers (*Figure 3A*). By 96 hr, the number of detectable apoptotic cells was comparably reduced in both genotypes.

Steatosis is often seen in xenobiotic-induced liver damage (*Malhi and Gores, 2008*). At 24 hr post DEN treatment, H&E-stained ROCK1wt and ROCK1nc liver sections exhibited comparable hepatocyte ballooning and mild microvesicular steatosis around the centrilobular regions. By 48 hr, there was evidence of necrosis and inflammation in the centrilobular region in both genotypes, with larger necrotic areas and more associated inflammatory foci in ROCK1nc livers relative to ROCK1wt livers. There were marked histopathological differences at 72 hr after DEN treatment; ROCK1nc livers had significantly larger areas of centrilobular necrosis, associated with inflammation, relative to ROCK1wt livers (*Figure 3B*). ROCK1nc livers also displayed extensive hepatocellular ballooning, micro and macrovesicular steatosis at the centrilobular zone. By 96 hr, livers from both genotypes exhibited reduced damage-associated lesions and necrotic areas associated with inflammatory foci. However, ROCK1nc livers continued to have significantly larger necrotic areas relative to ROCK1wt livers (*Figure 3B*). Consistent with the presence of damage that affected liver function, the metabolic enzyme alanine transaminase (ALT) (*Figure 3C*) and the heme catabolite bilirubin (*Figure 3D*) were elevated following DEN treatment, with significantly higher levels after 72 hr in ROCK1nc mice relative to ROCK1wt mice. By 96 hr, ALT and bilirubin levels were comparable in both genotypes (*Figure 3C, D*). The differences between the genotypes in the magnitudes and kinetics of liver damage (*Figure 3*) suggest that an additional process amplified the initial hepatotoxicity over time, culminating in significantly more tissue pathologies in ROCK1nc livers 72 hr after DEN administration. By 96 hr, livers from both ROCK1wt and ROCK1nc mice appeared to be undergoing comparable resolution of liver injury, indicating that the absence of ROCK1 activation in apoptotic cells did not affect recovery from the apoptotic stimulus.

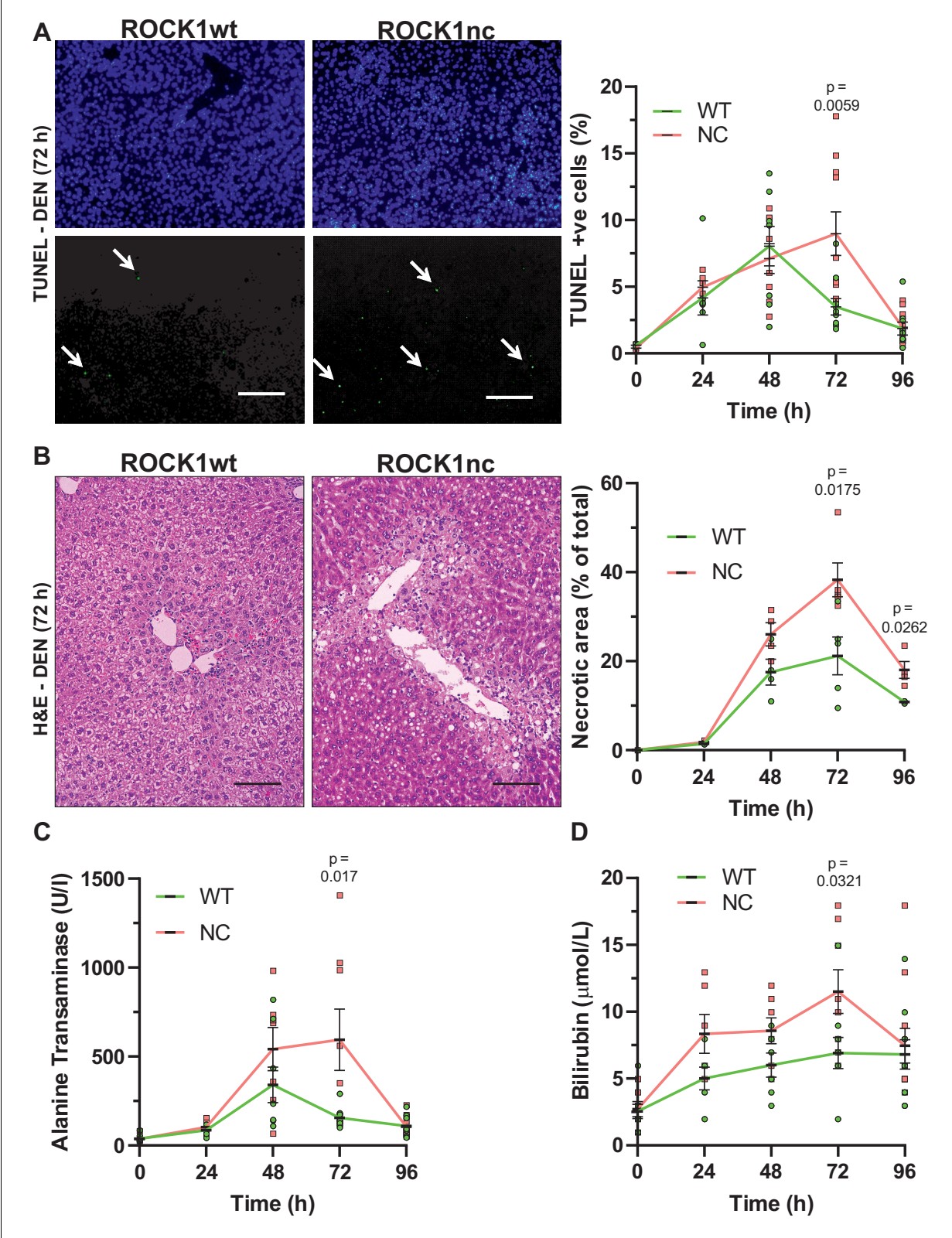

**Figure 3.** Elevated chemically induced liver damage in ROCK1nc mice. (**A**) Terminal deoxynucleotidyl transferase dUTP nick end labelling (TUNEL) staining of apoptotic cells in representative liver sections from ROCK1wt or ROCK1nc mice 72 hr after diethylnitrosamine (DEN) injection. TUNEL-positive cells (lower panel, green) were scored as a percentage of the total 4′,6-diamidino-2-phenylindole (DAPI; upper panel, blue) cell numbers from 10 random fields per section per mouse. Scale bar = 100 μm. Right panel: means ± SEM, n = 3–11 mice at each time. Pairwise Student's t-test at each

*Figure 3 continued on next page*

*Figure 3 continued*

time. (**B**) Haematoxylin and eosin (H&E) staining of representative liver sections from ROCK1wt or ROCK1nc mice 72 hr after DEN injection. Scale bar = 100 µm. Liver necrotic areas were scored as a percentage of the total area from 10 random fields per section per mouse. Right panel: means ± SEM, n = 3–5 mice at each time. Pairwise Student's t-test at each time. (**C**) Sera from 10-week-old homozygous ROCK1wt (WT) and ROCK1nc (NC) mice treated with DEN (100 mg/kg body weight) were assessed for alanine transaminase and (**D**) bilirubin levels at indicated times after intraperitoneal injection. Means ± SEM, n = 6–11 mice at each time. Pairwise Student's t-test at each time.

The online version of this article includes the following figure supplement(s) for figure 3:

**Figure supplement 1.** Absence of indications of autoimmunity in aged ROCK1wt or ROCK1nc kidneys or spleens.

**Figure supplement 2.** Ageing does not affect blood cells, liver or spleen masses differently in ROCK1wt or ROCK1nc mice.

Macrophages in liver sections were detected using the F4/80 monoclonal antibody (*Austyn and Gordon, 1981*) that recognizes both resident Kupfer cells and recruited monocyte-derived macrophages (*Guillot and Tacke, 2019*). Over the 4 days after DEN treatment, F4/80-positive cell numbers were comparably reduced in ROCK1wt and ROCK1nc livers (*Figure 4A*). In contrast, S100A9-positive neutrophils were present at significantly higher numbers in ROCK1nc livers 72 hr after DEN administration relative to ROCK1wt livers (*Figure 4B*). This difference in neutrophil recruitment to ROCK1nc livers was not replicated in an acute peritonitis model in which zymosan, a polysaccharide cell wall component derived from *Saccharomyces cerevisiae* (*Cash et al., 2009*), was injected into the peritoneal cavity of ROCK1wt and ROCK1nc mice to induce acute inflammation. At either 24 or 72 hr following zymosan injection, the numbers of neutrophils (*Figure 4C*), white blood cells (*Figure 4D*), monocytes (*Figure 4E*) or lymphocytes (*Figure 4F*) were equivalent in ROCK1wt and ROCK1nc mice. These results are consistent with the greater neutrophil numbers in ROCK1nc livers being a result of differences in the level of recruitment signal(s), rather than differing responses of neutrophils.

Although DAMP-induced sterile inflammation is an important mechanism for removing cellular debris and stimulating regeneration, aberrant levels of inflammation can provoke additional tissue injury (*Chen and Nuñez, 2010*). In fact, release of the HMGB1 DAMP protein from necrotic hepatocytes was shown to recruit neutrophils that amplified the original injury (*Huebener et al., 2015*). HMGB1 is a chromatin-associated protein that translocates from nucleus to cytoplasm, from which it may undergo extracellular release during cell stress or death (*Khambu et al., 2019*). Immunohistochemical (IHC) analysis 72 hr after DEN injection revealed significantly greater HMGB1 cytoplasmic translocation in ROCK1nc liver sections relative to ROCK1wt (*Figure 5A*). The ability of HMGB1 to recruit neutrophils to sites of injury stems from its interactions with DAMP receptors including TLR4 (*Sabroe et al., 2005*). Consistent with HMGB1-mediating neutrophil recruitment in ROCK1nc livers, HMGB1 inhibition with glycyrrhizin (GLZ), which binds directly with HMGB1 and antagonizes its chemoattractant activity (*Mollica et al., 2007*), or the TLR4 intracellular signalling inhibitor CLI095 (*Ii et al., 2006*) each significantly reduced neutrophil recruitment 72 hr after DEN treatment (*Figure 5B*). Inhibition of HMGB1 and TLR4 also led to significantly fewer TUNEL-positive cells (*Figure 5C*) and reduced serum ALT levels (*Figure 5D*). However, neither treatment affected the numbers of F4/80-stained macrophages (*Figure 5E*). These results are consistent with the interpretation that the ROCK1nc mutation shifts hepatocyte cell death towards a more necrotic-like form that results in greater HMGB1-TLR4-dependent neutrophil recruitment and consequent amplification of DEN-induced liver damage.

## Absence of ROCK1 cleavage reduces HCC tumours

In addition to inducing acute liver damage, DEN administration results in HCC associated with activating *Hras* mutations (*Chen et al., 1993*; *Buchmann et al., 1991*) and inactivating *Trp53* mutations (*Smith et al., 1991*). To determine whether the ROCK1nc mutation that amplified DEN-induced liver damage would affect HCC development, 14-day-old male ROCK1wt and ROCK1nc mice were injected IP with 400 µg DEN, then aged for 6 or 9 months (*Figure 6A*). The fewer number of liver tumours in ROCK1nc mice were clearly evident in macroscopic views of isolated livers (*Figure 6B*), which were confirmed by the significant differences in tumour numbers after 6 or 9 months determined by counting tumour numbers in H&E-stained ROCK1wt and ROCK1nc liver sections (*Figure 6C*). Despite the differences in tumour numbers, total tumour volumes per liver did not differ

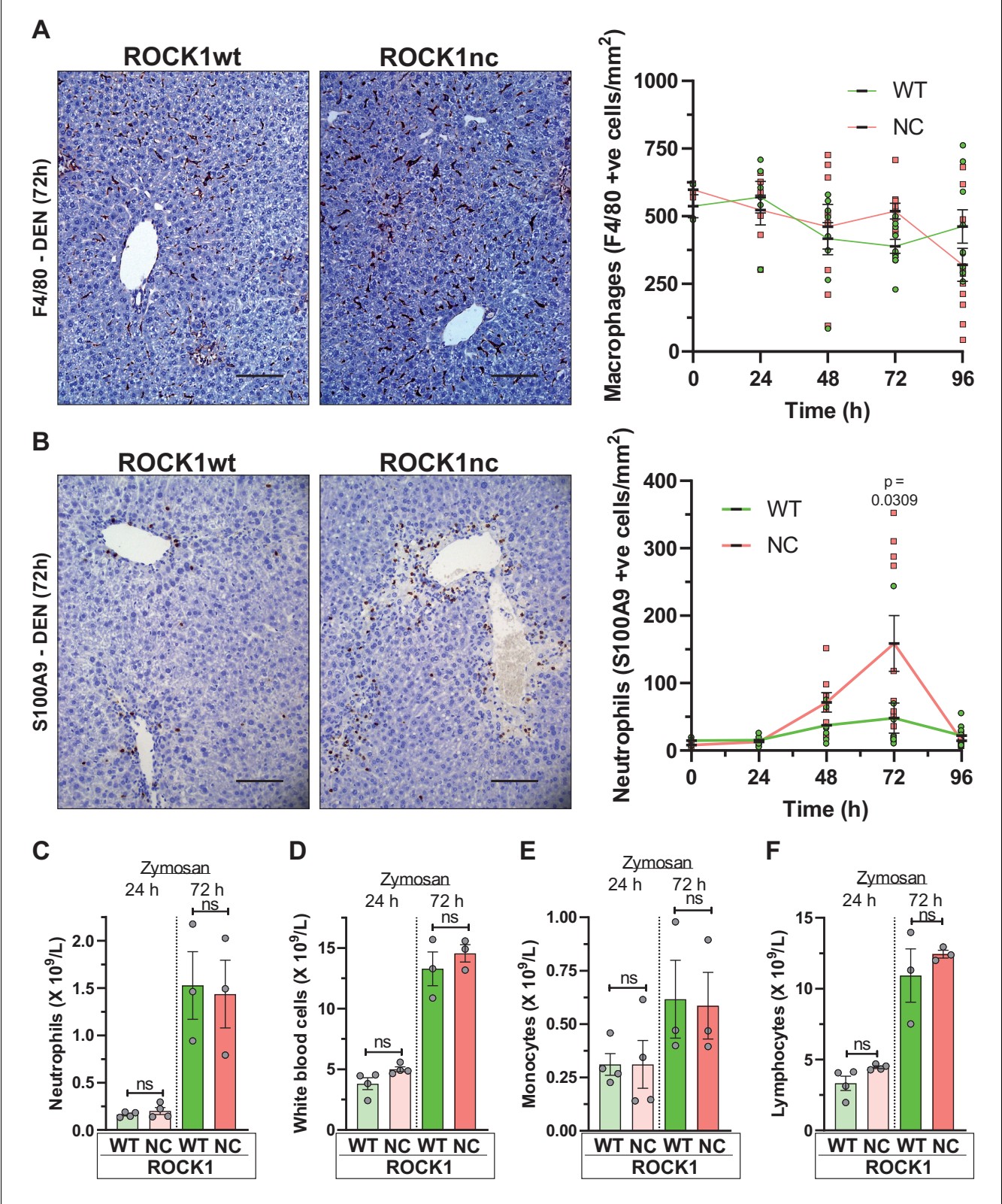

**Figure 4.** Higher neutrophil infiltration in chemically damaged ROCK1nc livers. (**A**) Immunohistochemical (IHC) staining with F4/80 antibody of formalin-fixed paraffin-embedded (FFPE) liver sections from homozygous ROCK1wt (WT) or ROCK1nc (NC) mice 72 hr after diethylnitrosamine (DEN) injection. Scale bar = 100 μm. F4/80-positive macrophages were scored as number per mm$^2$ in five random fields per section per mouse. Right panel: means ± SEM, n = 3–11 mice at each time. (**B**) IHC staining with anti-S100A9 antibody of FFPE liver sections from ROCK1wt or ROCK1nc mice 72 hr

*Figure 4 continued on next page*

*Figure 4 continued*

after DEN injection. Scale bar = 100 µm. S100A9-positive neutrophils were scored as number per mm$^2$ in five random fields per section per mouse. Right panel: means ± SEM, n = 3–11 mice at each time. Pairwise Student's t-test at each time. (C) Homozygous WT or NC mice were treated with 1 mg zymosan by intraperitoneal injection. At the indicated times after injection, the numbers of neutrophils, (D) white blood cells, (E) monocytes and (F) lymphocytes in the peritoneal cavity were determined. Boxes indicate minimum to maximum values with median designated by a line, n = 4 WT mice, 3 NC mice. Pairwise Student's t-test within times between genotypes; ns: not significant.

significantly between ROCK1wt and ROCK1nc mice at 6 or 9 months (*Figure 6D*). Interestingly, although there was a significant difference after 6 months in the number of lung metastases, this difference was not observed after 9 months (*Figure 6E*). Despite the large difference in tumour numbers, liver masses and liver/body mass ratios were not different between the genotypes (*Figure 7A–C*). Liver tumours were classified as described by *Yanagi et al., 1984* into the following grades: hepatic nodule 1 (HN1), in which nodules differed only slightly from the normal hepatic architecture but with obvious compression of surrounding parenchyma; hepatic nodule 2 (HN2), consisting of two- to three-cell thick irregular trabecular arrangements with well-maintained sinusoidal spaces; and HCC, which exhibited irregular macrotrabecular arrangement with dilated sinusoidal spaces (*Figure 7—figure supplement 1A–F*). Liver tumours from 12 ROCK1wt and 12 ROCK1nc mice indicated no significant differences in tumour grades (*Figure 7D*). Both genotypes displayed similar degrees of portal, perivenular (centrilobular zone) and pericellular fibrosis (extending along the sinusoids) as detected by Sirius Red staining of collagen (*Figure 7E*). Quantification of Sirius Red-stained collagen revealed no difference in the percentage of fibrotic area between ROCK1wt and ROCK1nc mice (*Figure 7E*). These results indicate that the principal difference between the liver tumours in ROCK1wt and ROCK1nc mice was related to their initiation rather than growth or progression.

Given that acute DEN treatment resulted in greater liver damage and hepatocyte cell death in ROCK1nc mice relative to ROCK1wt mice (*Figure 4*), a possibility was that more mutated hepatocytes, which might have given rise to liver tumours, were killed following DEN administration in ROCK1nc livers than in ROCK1wt livers. To test this hypothesis. ROCK1nc mice were treated with GLZ or DEN alone, or with the combination as shown in *Figure 8A* over the 4-day tumour initiation period, which recapitulates the GLZ dosing regimen that reduced neutrophil recruitment (*Figure 5B*), hepatocyte death (*Figure 5C*) and liver damage as determined by serum ALT levels (*Figure 5D*). After 9 months, during which there were no further treatments with either DEN or GLZ, the number of liver tumours induced by DEN plus GLZ was significantly higher than DEN alone, with GLZ treatment alone having no effect on tumour numbers (*Figure 8B*). There were no significant differences in the number of lung metastases between DEN or DEN plus GLZ-treated mice (*Figure 8C*), nor in the liver masses or liver/body mass ratios between three treatment conditions (*Figure 8—figure supplement 1A–C*). Taken together, these results indicate that the acute injury amplification associated with higher levels of neutrophil recruitment observed in DEN-treated ROCK1nc mice had a long-term benefit in reducing HCC tumour numbers (*Figure 8D*). Furthermore, the ability of GLZ to attenuate neutrophil recruitment in DEN-treated ROCK1nc livers was associated with more HCC tumours, resembling the situation in ROCK1wt livers (*Figure 8D*). Thus, the benefit from the ability of ROCK1 activation and consequent cell contraction to limit sterile inflammation and damage amplification during acute xenobiotic-induced tissue-scale cell death comes at the long-term cost of less efficient tumour suppression due to the survival of potential tumour-initiating cells.

## Discussion

The profound morphological events of cell and nuclear contraction, membrane blebbing and apoptotic body formation were the first defining characteristics associated with the form of cell death that was named 'apoptosis' by Kerr et al. in 1972. In fact, the dramatic contraction of apoptotic cells, and likely the formation of apoptotic bodies, was first identified in the livers of yellow fever patients by the Johns Hopkins Hospital pathologist *Councilman, 1890*. These eosin-stained 'acidophilic bodies' became known as Councilman bodies, which have been used for decades as a diagnostic feature of liver pathologies. Despite this long-standing awareness of the morphological changes that cells

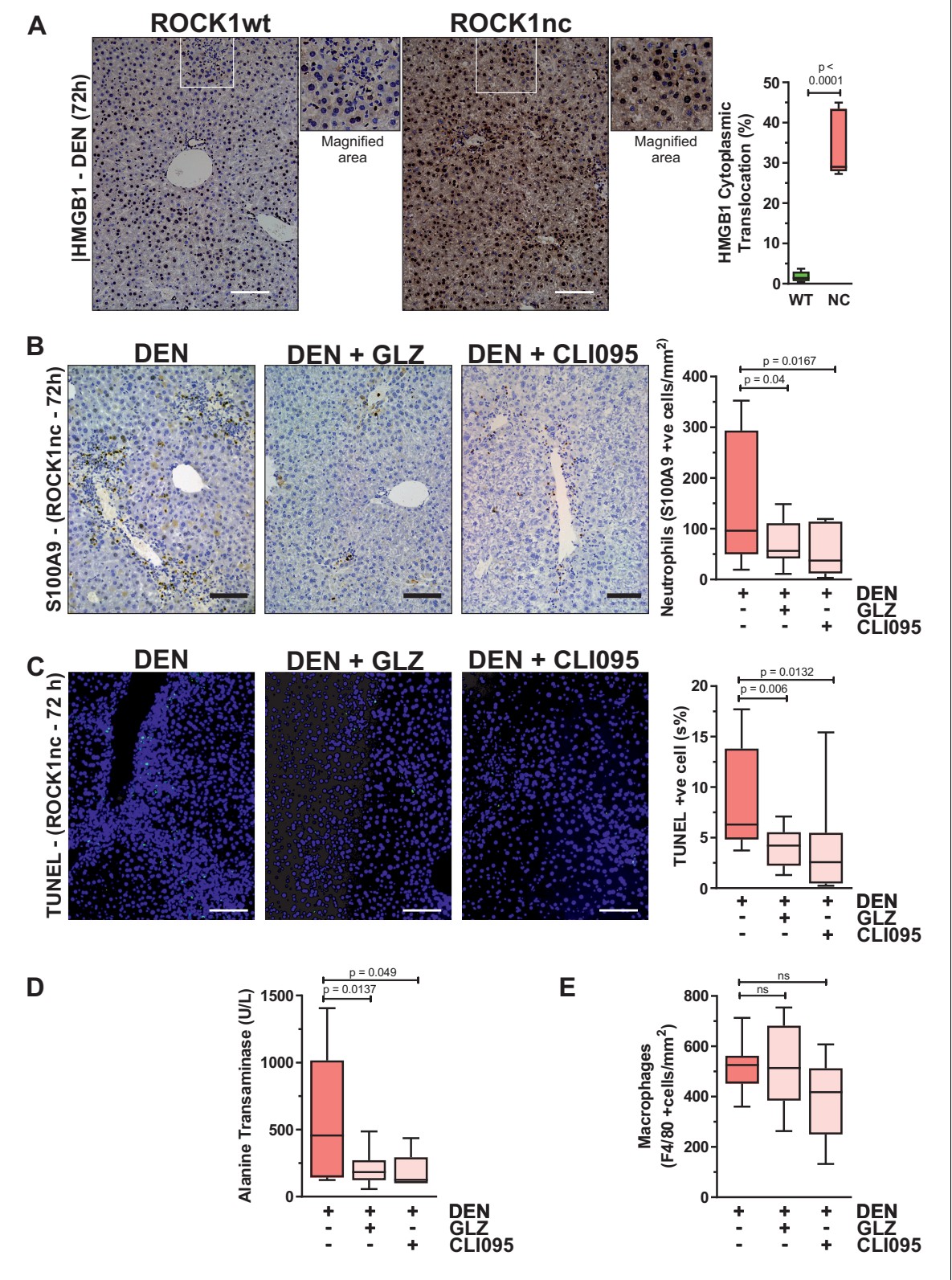

**Figure 5.** Inhibition of high-mobility group B1 (HMGB1) signalling reduces diethylnitrosamine (DEN)-induced liver damage. (**A**) Immunohistochemical (IHC) staining with anti-HMGB1 antibody of formalin-fixed paraffin-embedded (FFPE) liver sections from homozygous ROCK1wt (WT) or ROCK1nc (NC) mice 72 hr after DEN injection. White box indicates magnified area shown to the right of each image. Scale bar = 100 μm. Right panel: cells showing nucleus to cytoplasm translocation were scored as a percentage of the total cell numbers from five random fields per section per mouse. Boxes

*Figure 5 continued on next page*

*Figure 5 continued*

indicate upper and lower quartiles and median, whiskers indicate 5–95% percentiles, n = 5 mice. Student's t-test. (B) IHC staining with anti-S100A9 antibody of FFPE liver sections from ROCK1nc mice 72 hr after DEN injection alone or with glycyrrhizin (GLZ; 200 mg/kg) or CLI095 (3 mg/kg). Scale bar = 100 µm. S100A9-positive neutrophils 72 hr after DEN injection alone, or with GLZ or CLI095 (n = 10 mice for each condition), were scored as number per mm$^2$ in five random fields per section per mouse. Right panel: boxes indicate upper and lower quartiles and median, whiskers indicate 5–95% percentiles, pairwise one-tailed Student's t-test as indicated. (C) Terminal deoxynucleotidyl transferase dUTP nick end labelling (TUNEL) staining of apoptotic cells in representative liver sections from ROCK1nc mice 72 hr after DEN injection alone, or with GLZ or CLI095. Right panel: TUNEL-positive cells (green) were scored as a percentage of the DAPI-positive cells (blue) from 10 random fields per section per mouse following DEN injection alone or with GLZ or CLI095 (n = 10 mice for each condition). Scale bar = 100 µm. Right panel: boxes indicate upper and lower quartiles and median, whiskers indicate 5–95% percentiles, pairwise one-tailed Student's t-test as indicated. (D) Serum alanine transaminase levels in ROCK1nc mice 72 hr after DEN injection alone (n = 8 mice) or with GLZ (n = 10 mice) or CLI095 (n = 5 mice). Boxes indicate upper and lower quartiles and median, whiskers indicate 5–95% percentiles, pairwise one-tailed Student's t-test as indicated. (E) F4/80-positive macrophages in ROCK1nc mouse livers 72 hr after DEN injection alone or with GLZ or CLI095 (n = 10 mice for each condition), scored by IHC as number per mm$^2$ in five random fields per section per mouse. Boxes indicate upper and lower quartiles and median, whiskers indicate 5–95% percentiles, pairwise one-tailed Student's t-test as indicated; ns: not significant.

undergo during apoptotic death, their physiological roles in tissue homeostasis and organismal health have remained unclear. Cell-based experiments suggested that the forceful breaking of cells into pieces was important for efficient clearance by macrophages (*Orlando and Pittman, 2006*). In addition, apoptotic cell death was proposed to limit inflammatory responses due to cellular contents being enclosed in membranes, in contrast to the robust inflammation induced by necrotic cell death (*Kerr et al., 1972*). Despite the prevalence of these concepts, there had not been a direct in vivo model system that allowed for an evaluation of the importance of apoptotic cell morphologies.

The universal conservation of the ROCK1 second aspartate residue (D1113 in humans) within the caspase cleavage sequence in vertebrate species (*Table 1*) is suggestive of caspase activation of ROCK1 being important such that it was conserved through evolution. Additional regulators of actin cytoskeleton organization have been reported both as being caspase cleaved and influencing apoptotic cell morphologies, including p21-activated kinase 2 (PAK2) (*Rudel and Bokoch, 1997*; *Lee et al., 1997*), Lim kinase 1 (LIMK1) (*Tomiyoshi et al., 2004*) and the myosin phosphatase targeting subunit 1 (MYPT1) of the myosin phosphatase complex (*Iwasaki et al., 2013*). As a result, despite the ability of ROCK inhibitors to block apoptotic cell contraction, nuclear disruption and membrane blebbing (*Coleman et al., 2001*; *Croft et al., 2005*; *Wickman et al., 2013*), the primacy of caspase-mediated ROCK1 activation as the principal driver of apoptotic cell morphologies was previously unclear.

Mutation of the ROCK1 caspase-cleavage site did not affect basal ROCK1 activity in vitro (*Figure 1A*) or in cells (*Figure 1B*), but did make the protein resistant to caspase-mediated activation (*Figure 1A*) and prevented the increased MLC phosphorylation typically observed in apoptotic cells (*Figure 1C*). Importantly, this single amino acid change blocked the ability of apoptotic cells to generate contractile force (*Figure 2E*) and altered the morphology of cells undergoing apoptotic death such that the typical pattern of contraction and membrane blebbing was shifted to a necrosis-like form of cell death (*Figure 2A*) that was more prone to the release of cellular contents over time (*Figure 1F*). These results demonstrate the critical and central role of caspase-mediated ROCK1 activation in the manifestation of typical apoptotic cell morphological changes.

Despite the presumed importance of proper morphological events during apoptotic cell death, homozygous ROCK1nc mice were apparently healthy and reproduced normally over multiple rounds. Crosses of ROCK1wt/nc heterozygous breeding pairs resulted in offspring at expected Mendelian ratios (*Figure 1—figure supplement 2D*). Adult mice aged for up to a year showed no overt signs of autoimmunity (e.g. *Figure 3—figure supplements 1* and *2*), indicating that the ROCK1nc mice tolerated any effects that the ROCK1nc mutation might have had on apoptotic cell death during the normal cell turnover in developing and adult tissues that occurs at a low but constitutive rate. It remains an open possibility that autoimmunity could spontaneously develop with low penetrance, with or without prolonged ageing, which would require very large mouse cohorts to detect. To determine whether typical morphological effects were important in response to a strong apoptotic stimulus, mice were treated with DEN, which acts in a liver-selective manner due to the need to be metabolized to a reactive compound by the cytochrome P450 enzyme CYP2E1 (*Kang et al., 2007*; *Bakiri and Wagner, 2013*). In this model of tissue-scale-induced apoptosis, the ROCK1nc mutation

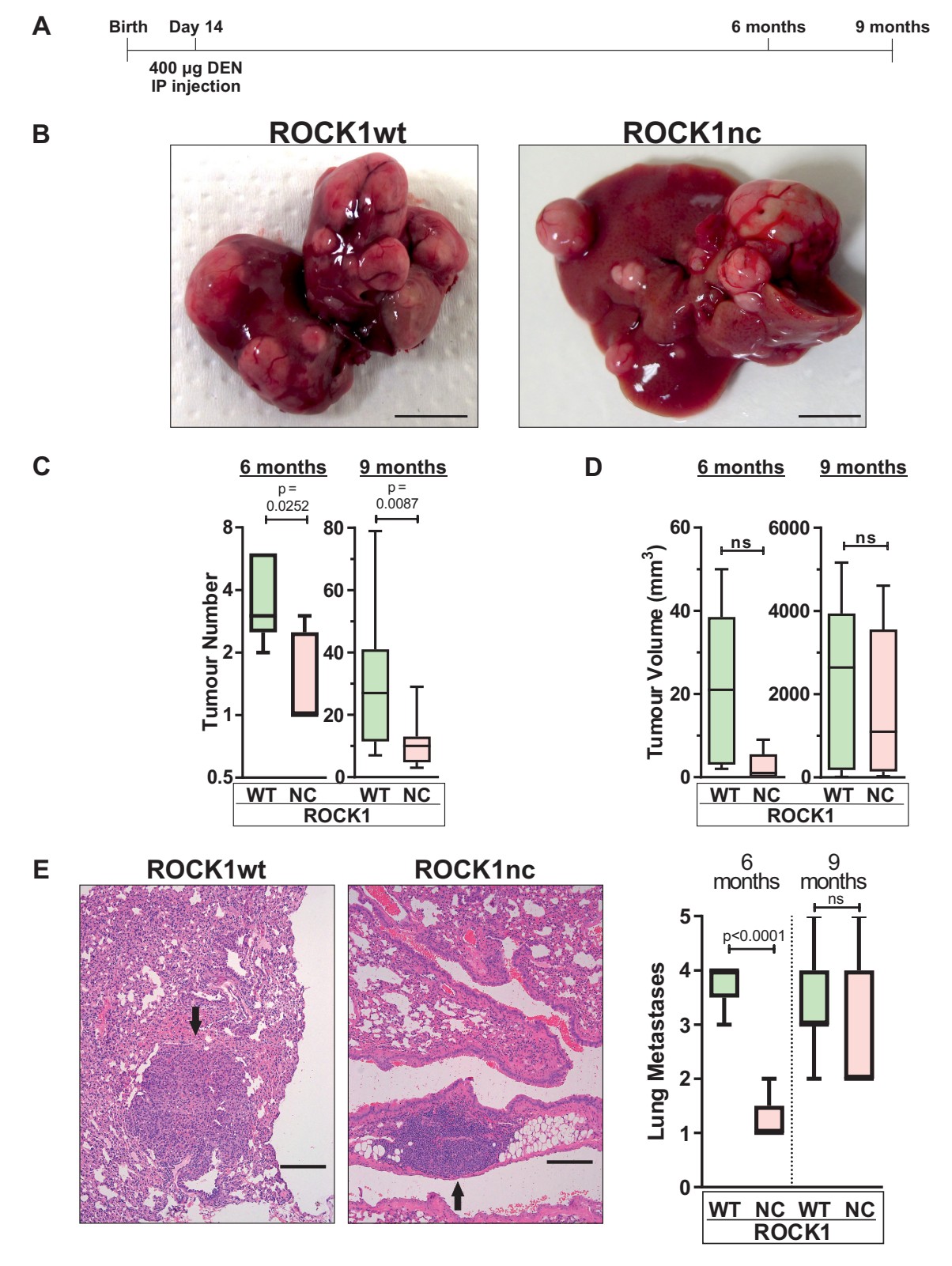

**Figure 6.** Diethylnitrosamine (DEN)-induced hepatocellular carcinoma (HCC) tumour numbers are lower in ROCK1nc mice. (**A**) Timeline of DEN-induced HCC protocol. (**B**) Macroscopic view of representative livers from ROCK1wt and ROCK1nc mice 9 months after HCC initiation with DEN. Scale bars = 1 cm. (**C**) Tumour numbers were counted in ROCK1wt and ROCK1nc livers 6 months (n = 5 mice for each genotype) or 9 months (12 mice for each genotype) after HCC initiation with DEN. Boxes indicate upper and lower quartiles and median, whiskers indicate 5–95% percentiles, pairwise Student's

*Figure 6 continued on next page*

Figure 6 continued

t-test as indicated. (D) Total tumour volume per ROCK1wt or ROCK1nc liver 6 months (n = 5 mice for each genotype) or 9 months (12 mice for each genotype) after HCC initiation with DEN. Boxes indicate upper and lower quartiles and median, whiskers indicate 5–95% percentiles, pairwise Student's t-test as indicated; ns: not significant. (E) Representative H&E-stained lung sections from ROCK1wt and ROCK1nc mice 9 months after HCC initiation with DEN. Black arrow indicates metastatic nodules. Scale bars = 100 μm. Numbers of lung metastases were counted in ROCK1wt and ROCK1nc mice 6 months (n = 5 mice for each genotype) or 9 months (12 mice for ROCK1wt, 7 mice for ROCK1nc) after HCC initiation with DEN. Right panel: boxes indicate upper and lower quartiles and median, whiskers indicate 5–95% percentiles, pairwise Student's t-test as indicated; ns: not significant.

amplified the initial DEN-induced liver damage, which could be significantly reduced by inhibition of HMGB1-TLR4 signalling, key mediators of the pathogen-free sterile inflammatory response (*Chen and Nuñez, 2010*). These observations are the first indication of a physiological role for ROCK1 activation by caspase cleavage, and the apoptosis-associated morphological events, in the maintenance of tissue homeostasis.

Chronic inflammation has well-documented roles in positively contributing to all stages of tumour initiation, growth and progression (*Greten and Grivennikov, 2019*). However, the lower number of liver tumours in ROCK1nc mice relative to ROCK1wt mice reveals a significant role of acute sterile inflammation as a mechanism for tumour suppression. This function may be particularly apropos in the liver where xenobiotic metabolism by P450 enzymes may inadvertently convert relatively inert compounds into reactive molecules with toxic and mutagenic actions (*Wogan et al., 2004*). Although the majority of cells exposed to these agents would likely be so damaged as to induce apoptosis, a subpopulation with altered genomic DNA might survive and replicate, thus passing on their acquired mutations that could eventually contribute to tumour initiation (*Schulte-Hermann et al., 1999*). In this context, neutrophil-mediated injury amplification would serve a useful purpose; the collateral damage they inflict in proximity to cells that died as a consequence of exposure to toxic mutagens would also eliminate many potential liver cancer-initiating cells (*Li and Zhu, 2019*). A fine balance between the level of injury amplification that can be tolerated and the benefits from long-term tumour suppression has likely been achieved through evolution, including a role for ROCK1 cleavage to limit the extent of sterile inflammation.

Previous studies have evaluated the contribution of key apoptosis effector proteins by deleting them in genetically modified mice, such as deletion of the caspase-activated DNAse (CAD), which revealed a role for apoptotic DNA degradation in thymic development (*Kawane et al., 2003*). There are very few examples in which the importance of caspase cleavage of target proteins has been evaluated in vivo. A recent report showed that a D325A mutation in the caspase 8 cleavage site of the receptor-interacting serine/threonine-protein kinase 1 (RIPK1) resulted in embryonic lethality in homozygous mutant mice that was associated with hypersensitity to TNFα (*Lalaoui et al., 2020*). This study revealed a role for caspase activation of ROCK1 in liver homeostasis when challenged with a strong apoptotic stimulus and a previously unknown role for sterile inflammation as a mechanism of tumour suppression. Future studies will likely reveal additional important contributions of the proper development of apoptotic cell morphological changes to the health of other tissues and organs.

## Materials and methods

### In vitro ROCK1 caspase cleavage and kinase assay

ROCK1 kinase assays were performed with immunopurified full-length Myc-tagged wild-type, D1113A, KD K105M and CA Δ2 ROCK1 plasmids (*Coleman et al., 2001*; *Ishizaki et al., 1997*) that had been expressed in HEK293T cells. Confluent HEK293T human embryonic kidney cells grown in DMEN with 10% FBS were transfected with 10 μg of plasmid with Fugene (Promega) in 10 cm plates and incubated at 37°C for 48 hr. Transfected cells were washed, collected and lysed with kinase lysis buffer (50 mM Tris pH 7.5, 150 mM NaCl, 1 mM EDTA, 1% (v/v) Triton X-100, 20 mM NaF, 0.2 mM $Na_3VO_4$, 2 nM β-glycerolphosphate, 1X cOmplete protease inhibitor cocktail, 1 μg/mL aprotinin, 1 mM phenylmethylsulfonyl fluoride [PMSF]). Anti-Myc-tag antibody was added to approximately 450 μg of cellular lysate overnight at 4°C. Protein bound to antibody was precipitated with protein G Sepharose and washed in buffer (50 mM Tris pH 7.5, 150 mM NaCl), then re-suspended in 100 μL of

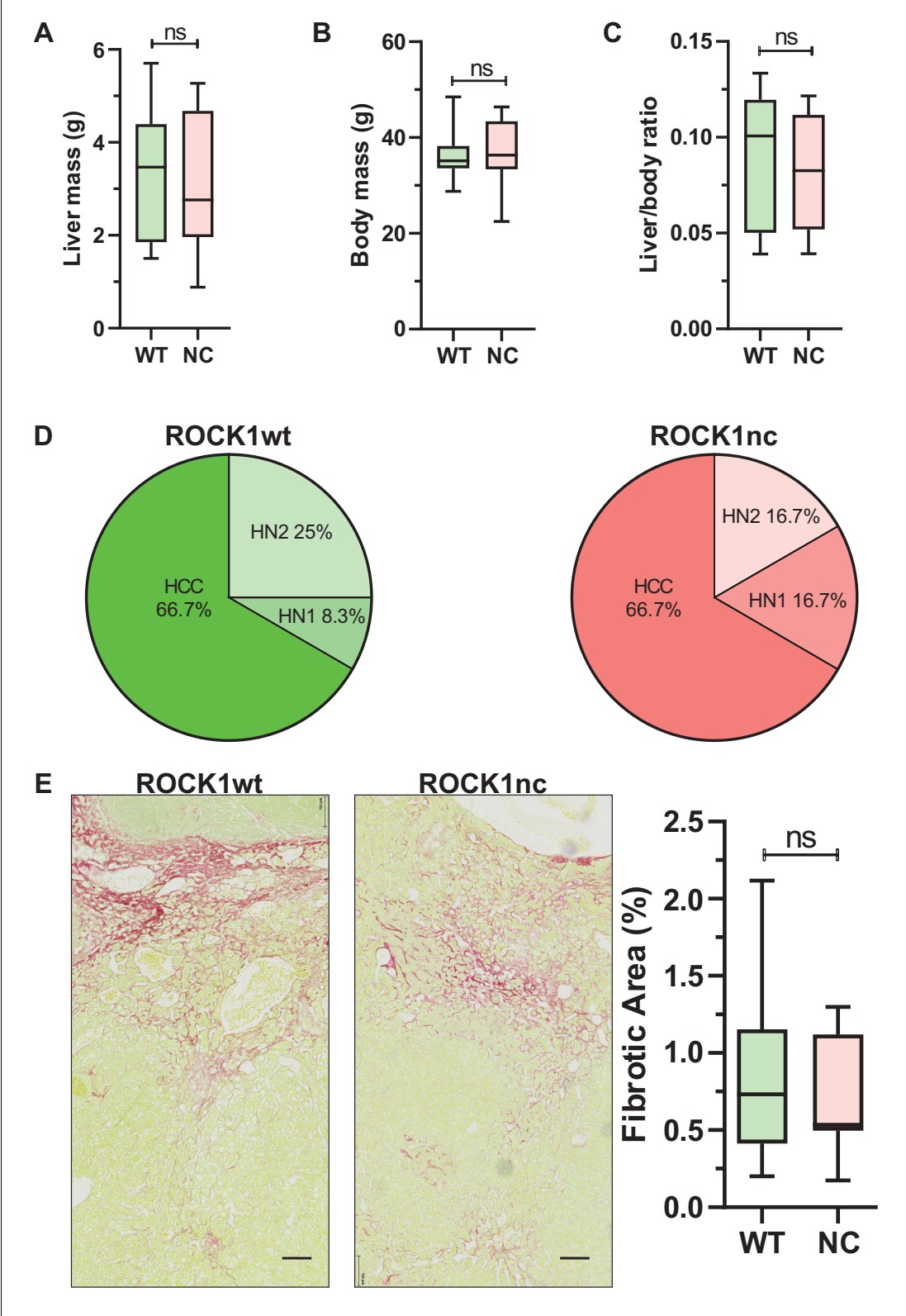

**Figure 7.** Similar liver mass and tumour grades in diethylnitrosamine (DEN)-treated ROCK1wt and ROCK1nc mice. (**A**) Liver mass, (**B**) body mass and (**C**) liver to body ratios for ROCK1wt and ROCK1nc mice (n = 12 mice for each genotype) 9 months after hepatocellular carcinoma (HCC) initiation with DEN. Boxes indicate upper and lower quartiles and median, whiskers indicate 5–95% percentiles, pairwise Student's t-test as indicated; ns: not significant. (**D**) Tumour classification in ROCK1wt (left) and ROCK1nc (right) mice (n = 12 mice for each genotype) 9 months after HCC initiation with

*Figure 7 continued on next page*

*Figure 7 continued*

DEN. H&E-stained sections liver tumours were classified into three types: hepatic nodule 1 (HN1), hepatic nodule 2 (HN2) and HCC in increasing order of tumour grade as described by *Yanagi et al., 1984*. HN1 mainly comprised tumour nodules with slight difference from the normal hepatic architecture but produced obvious compression of the surrounding parenchyma. HN2 consisted of two- to three-cell thick irregular trabecular arrangements with well-maintained sinusoidal spaces. HCC exhibited irregular macrotrabecular arrangement with dilated sinusoidal spaces. (E) Photomicrographs of liver sections from ROCK1wt and ROCK1nc mice 9 months after DEN injection stained with Sirius Red and counterstained with Fast Green. Scale bars represent 100 μm. Percentage fibrotic area measured as Sirius Red positive area relative to the total area analysed, showing no significant difference between the two genotypes (ROCK1wt n = 8 mice; ROCK1nc n = 7 mice). Right panel: boxes indicate upper and lower quartiles and median, whiskers indicate 5–95% percentiles, pairwise Student's t-test as indicated; ns: not significant.

The online version of this article includes the following figure supplement(s) for figure 7:

**Figure supplement 1.** Histopathological features of diethylnitrosamine (DEN)-induced hepatocellular carcinoma (HCC).

20 mM PIPES pH 7.2, 100 mM NaCl, 10 mM DTT, 1 mM EDTA, 0.1% (v/v) CHAPS, 10% (w/v) sucrose. Active caspase 3 (Sigma) was added to tubes, then incubated at 37°C for 1 hr prior to kinase assay. Beads were washed 3× in 50 mM Tris pH 7.5, 150 mM NaCl, then were re-suspended in 94 μL of kinase buffer (20 mM Tris HCl pH 7.4, 0.5 mM MgCl$_2$, 0.01% Tween 20 and 1 mM DTT) with 2 μL of 5 mM ATP (100 μM) and 4 μL of recombinant MLC and incubated with constant agitation at 30°C for 1 hr. To stop the reaction, 100 μL of boiling 2% Sodium dodecyl sulfate (SDS) was added to the samples and the reactions were incubated for 10 min at 70°C. The samples were centrifuged at 3600 rpm for 2 min and supernatants were collected for western blots. Samples were separated on a 10% SDS-PAGE gel, then transferred to nitrocellulose and blocked with bovine serum albumin (BSA). Membranes were probed for Myc-tag (Cell Signaling), MLC (Santa Cruz) or pMLC (Cell Signaling). Antibody binding was visualized with fluorescent secondary antibodies using an Odyssey LiCOR scanner.

## Targeting vector generation

The 5.1 kb 3′ homology arm for targeting *Rock1* was isolated from a BAC clone containing the murine *Rock1* gene by gap repair recombineering in DY380 cells as described in *Lee et al., 2001*. BAC clones were from the Sanger AB2.2 mouse genomic DNA library (representative of the mouse 129 strain) and identified as 107N12, 208K1 and 459E7.

A retrieval vector was generated by PCR-amplifying short (approximately 500 bp) arms that represented the extreme ends of the 5.1 kb 3′ homology arm and sub-cloning these into pCR-blunt 2.1 (Invitrogen) such that the adjacent sequences were separated by an AscI site to allow linearization of the vector between the amplified arms.

DY380 cells carrying the target BAC were grown at 30°C to an OD$_{600}$0.5–0.6, then shifted to 42°C for 15 min to heat-shock induce lambda phage recombination proteins that drive the retrieval. Heat-shocked cells were harvested and washed twice in sterile distilled H$_2$O. Pelleted cells (25 μL) were resuspended in residual H$_2$O, then mixed with 50–100 ng (5 μL volume max) of the gel-purified, *AscI*-linearized retrieval vector. Following electroporation, bacterial cells were allowed to recover by shaking at 30°C in 900 μL Super Optimal broth with Catabolite repression (SOC), then plated on LB-agar plates supplemented with ampicillin.

The 3.1 kb 5′ homology arm of *Rock1* was generated from BAC DNA by high-fidelity PCR amplification. An amino acid point mutation to make ROCK1 caspase-resistant was introduced into exon 27 of the *Rock1* gene using a mutated oligonucleotide primer with a Stratagene QuickChange II XL site-directed mutagenesis kit.

The mutated plasmid was sequenced to confirm the mutation (which generated a novel *PstI* restriction site) and the conservation of other exons within the fragment. In sequencing, a number of polymorphisms were identified in the 129 strain-derived genomic DNA (confirmed by sequencing from independent PCR products from tail DNA) relative to the C57Bl/6 DNA sequence archived by Ensemble. Two changes were identified within exon 27 and were conservative with reference to the encoded amino acid sequence. The 5′ and mutated 3′ arms were cloned directionally into pDupDel Neo (gift of Prof. Oliver Smithies) using restriction sites introduced during PCR cloning and recombineering, respectively. This placed the mutated exon 27 proximal to the selection cassette to decrease the chance of loss of the mutation during subsequent homologous recombination in mouse embryonic stem cells (mESCs). Since pDupDel Neo contains a high-activity form of the

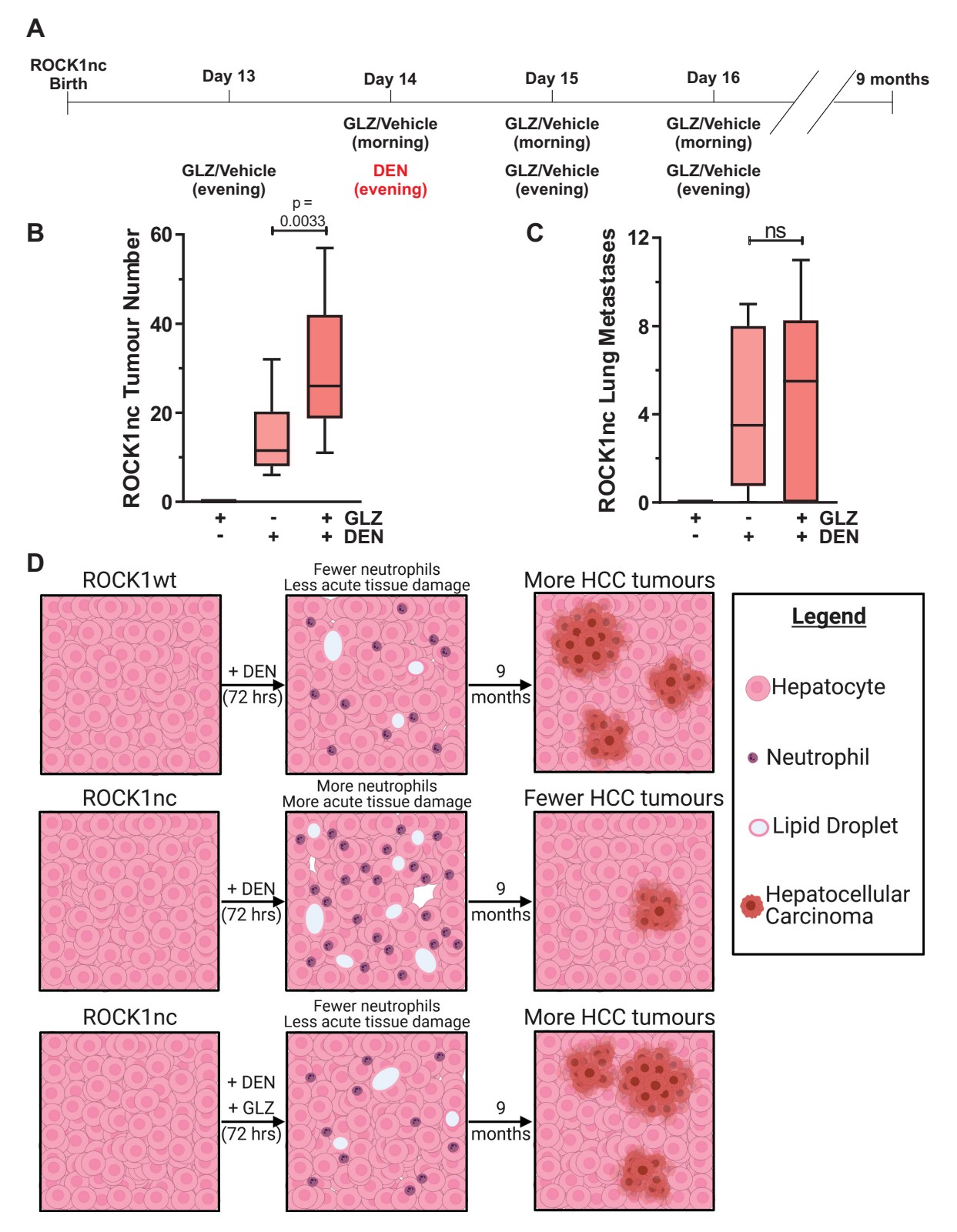

**Figure 8.** Glycyrrhizin (GLZ) increases diethylnitrosamine (DEN)-induced liver tumour number in ROCK1nc mice. (**A**) Timeline of the DEN-induced hepatocellular carcinoma (HCC) and GLZ treatment protocol. (**B**) The number of liver tumours in ROCK1nc mice 9 months after treatment with GLZ alone (n = 10 mice) or after HCC initiation with DEN alone (n = 14 mice) or with GLZ (n = 10 mice). Boxes indicate upper and lower quartiles and median, whiskers indicate 5–95% percentiles, pairwise Student's t-test as indicated. (**C**) Numbers of lung metastases in ROCK1nc mice 9 months after

*Figure 8 continued on next page*

*Figure 8 continued*

treatment with GLZ alone (n = 10 mice) or after HCC initiation with DEN alone (n = 14 mice) or with GLZ (n = 10 mice). Boxes indicate upper and lower quartiles and median, whiskers indicate 5–95% percentiles, pairwise Student's t-test as indicated; ns: not significant. (D) Schematic diagram depicting how acute acting neutrophils influence tumour development. ROCK1wt mice had fewer recruited neutrophils and less tissue damage than ROCK1nc mice 3 days after equivalent DEN treatment. The healthier ROCK1wt liver tissue enabled more DEN-mutated hepatocytes to initiate the HCC tumours that were observed 9 months later. Following GLZ administration, ROCK1nc mice resembled ROCK1wt mice with fewer neutrophils and less tissue damage acutely, and more HCC tumours 9 months later. Figure created with BioRender.com.
The online version of this article includes the following figure supplement(s) for figure 8:

**Figure supplement 1.** Similar liver mass in ROCK1nc mice treated with diethylnitrosamine (DEN) and glycyrrhizin (GLZ).

neomycin resistance gene, the original selection cassette was removed by Cre deletion in EL350 cells (*Lee et al., 2001*) and a low-activity version (expressed from a PGK/EM7 hybrid promoter) from pL452 (*Liu et al., 2003*) was inserted by the reverse process and selection on kanamycin. The resultant targeting vector (*Figure 1—figure supplement 2A*) was linearized with *NotI* restriction enzyme in preparation for transfection of mESCs.

## mESC vector transfection

G4 murine embryonic stem cells (gift of Andras Nagy and Marina Gertsenstein; *George et al., 2007*) were grown in ES medium Dulbecco's Modified Eagle Medium (DMEM [high glucose], 15% FBS, 2 mM glutamax, 0.1 mM non-essential amino acids, 0.1 mM β-mercaptoethanol 1000 U/mL ESGRO [Millipore], 33 µg/mL [gentamicin]) on MEF feeder layers. Cells were harvested from two T175 flasks 24 hr after their final split and plated for 20 min to remove the majority of MEFs. Cells were pelleted, washed once with PBS and counted.

Following resuspension in room temperature Embryomax transfection buffer (Millipore) at $1 \times 10^7$ cells/mL, five volumes of $8 \times 10^6$ cells were mixed with 40 µg of linearized targeting vector and the mixtures transferred to 4 mm electroporation cuvettes (BioRad) before electroporation at 250 V, 500 µF and infinite resistance. After recovery at room temperature for 20 min, cells were washed out of the cuvettes with regular ES medium and the contents of each cuvette split across four 10 cm plates of MEFs. After 24 hr, medium was changed to ES medium supplemented with 240 µg/mL G418. The plates were maintained in this medium (with daily changes) for 6 days.

Colonies selected for G418 resistance were plated onto five 96-well plates of MEFs and grown under G418 selection for 3 days. Plates were then archived for subsequent recovery and cells grown on parallel gelatinized plates. 96-well plates containing expanded ESC colonies were washed twice with 200 µL PBS per well and 50 µL of trypsin added. Following 5 min at 37°C, cells were pipetted up and down, then 50 µL of ES medium was added to terminate trypsinization. A 50 µL volume of trypsinized cells was transferred to a parallel round-bottomed 96-well plate containing 50 µL of 2× freezing medium (40% FBS, 40% ES medium, 20% dimethylsulfoxide [DMSO]) to act as a master plate. The plate was overlaid with 100 µL of embryo-tested mineral oil, sealed and frozen in a polystyrene box at −80°C before transfer to liquid nitrogen storage. The remaining trypsinized cells were transferred to gelatinized plates prepared by applying: 50 µL of 0.1% (w/v) porcine gelatin in PBS to each well of a 96-well plate and allowing to stand at room temperature for 20 min. After this time, the plate was drained and 50 µL of ES medium added. Cells from the trypsinized 96-well plate were transferred to the parallel gelatinized plate with additional ES medium (up to a final volume of 250 µL) and allowed to grow for 4 days upon which time the medium was yellowed. The plates were then drained and washed twice with PBS. 50 µL of lysis buffer (10 mM Tris HCl, pH 7.5, 10 mM EDTA, 10 mM NaCl, 1 mg/mL proteinase K, 0.5% (w/v) SDS) was added to each well, the plates sealed with adhesive sheets and incubated in a humidified chamber at 55°C overnight. Genomic DNA was precipitated by the addition of 100 µL of ice-cold ethanol-salt mix (75 mM NaCl in 100% ethanol) to each well. Plates were allowed to stand for a minimum of 1 hr at room temperature before being drained by careful inversion onto paper towels. The plate was then washed three times with 200 µL/well of 70% ethanol and allowed to air-dry for 20–30 min.

**Table 1.** Conservation of ROCK1 caspase cleavage site.

**Mammals**

| | |
|---|---|
| *Homo sapiens* | QLRAKLLDLSDSTSVASFPSA**DETD**GNLP |
| Chimp | QLRAKLLDLSDSTSVASFPSA**DETD**GNLP |
| Gorilla | QLRAKLLDLSDSTSVASFPSA**DETD**GNLP |
| Orangutan | QLRAKLLDLSDSTSVASFPSA**DETD**GNLP |
| Gibbon | QLRAKLLDLSDSTSVASFPSA**DETD**GNLP |
| Rhesus | QLRAKLLDLSDSTSVASFPSA**DETD**GNLP |
| Crab-eating macaque | QLRAKLLDLSDSTSVASFPSA**DETD**GNLP |
| Baboon | QLRAKLLDLSDSTSVASFPSA**DETD**GNLP |
| Green monkey | QLRAKLLDLSDSTSVASFPSA**DETD**GNLP |
| Marmoset | QLRAKLLDLSDSTSVASFPSA**DETD**GNLP |
| Squirrel monkey | QLRAKLLDLSDSTSVASFPSA**DETD**GNLP |
| Bushbaby | QLRAKLLDLSDSTSVASFPSA**DETD**GNLP |
| Chinese tree shrew | QLRAKLLDLSDSTSVASFPSA**DETD**GNLP |
| Squirrel | QLRAKLLDLSDSTSVASFPSA**DETD**GNLP |
| Lesser Egyptian jerboa | QLRAKLLDLSDSTSVASFPSA**DETD**GSLP |
| Prairie vole | QLRAKLLDLSDSTSVASFPSA**DETD**GNLP |
| Chinese hamster | QLRAKLLDLSDSTSIASFPSA**DETD**GNLP |
| Golden hamster | QLRAKLLDLSDSTSVASFPSA**DETD**GNLP |
| Mouse | QLRAKLLDLSDSTSVASFPSA**DETD**GNLP |
| Rat | QLRAKLLDLSDSTSVASFPSA**DETD**GNLP |
| Naked mole-rat | QLRAKLLDLSDSTSVASFPSA**DETD**GNLP |
| Guinea pig | QLRAKLLDLSDSTSVASFPSA**DETD**GNLP |
| Chinchilla | QLRAKLLDLSDSTSVASFPSA**DETD**GNLP |
| Brush-tailed rat | QLRAKLLDLSDSTSVASFPSA**DETD**GNLP |
| Rabbit | QLRAKLSDLSDSTSVASFPSA**DETD**GNLP |
| Pika | QLRAKLLDLSDSTSVASFPST**DE**a**D**GNLP |
| Pig | QLRAKLLDLSDSTSVASFPSA**DETD**GNLP |
| Alpaca | QLRAKLLDLSDSTSVASFPSA**DETD**GNLP |
| Bactrian camel | QLRAKLLDLSDSTSVASFPSA**DETD**GNLP |
| Dolphin | QLRAKLLDLSDSTSVASFPSA**DETD**GNLP |
| Killer whale | QLRAKLLDLSDSTSVASFPSA**DETD**GNLP |
| Tibetan antelope | QLRAKLLDLSDSTSVASFPSA**DETD**GNLP |
| Cow | QLRAKLLDLSDSTSVASFPSA**DETD**GNLP |
| Sheep | QLRAKLLDLSDSTSVASFPSA**DETD**GNLP |
| Domestic goat | QLRAKLLDLSDSTSVASFPSA**DETD**GNLS |
| Horse | QLRAKLLDLSDSTSVASFPSA**DETD**GNLP |
| White rhinoceros | QLRAKLLDLSDSTSVASFPSA**DETD**GNLP |
| Cat | QLRAKLLDLSDSTSVASFPSA**DETD**GNLP |
| Dog | QLRAKLLDLSDSTSIASFPSA**DETD**GNLP |
| Ferret | QLRAKLLDLSDSTSVASFPSA**DETD**GNLP |
| Panda | QLRAKLLDLSDSTSVASFPSA**DETD**GNLP |
| Pacific walrus | QLRAKLLDLSDSTSVASFPSA**DETD**GNLP |
| Weddell seal | QLRAKLLDLSDSTSVASFPSA**DETD**GNLP |
| Black flying-fox | QLRAKLLDLSDSTSVASFPSA**DETD**GNLP |

*Table 1 continued on next page*

| | |
|---|---|
| Megabat | QLRAKLLDLSDSTSVASFPSA**DETD**GNLP |
| David's myotis (bat) | QLRAKLLDLSDSTSVASFPSA**DETD**GNLP |
| Microbat | QLRAKLLDLSDSTSVASFPSA**DETD**GNLP |
| Big brown bat | QLRAKLLDLSDSTSVASFPSA**DETD**GNLP |
| Hedgehog | QLRAKLLDLADSTSVASFPSA**DETD**GNLP |
| Shrew | QLRSKLLDLSDSTSVASFPSA**DETD**GTLP |
| Star-nosed mole | QLRAKLLDLSDSTSVASFPSA**DETD**GNLP |
| Elephant | QLRAKLLDLSDSTSVASFPSA**DETD**GNLP |
| Cape elephant shrew | QLRAKLLDLSDSTSVASFPSA**DETD**GSLP |
| Manatee | QLRAKLLDLSDSTSVASFPSA**DETD**GNLP |
| Cape golden mole | QLRAKLLDLSDSTSVASFPSA**DETD**GNLP |
| Tenrec | QLRAKLLDLSDSTSVASFPSA**DETD**GNLP |
| Aardvark | QLRAKLLDLSDSTSVASFPSA**DETD**GNLP |
| Armadillo | QLRAKLLDLSDSTSVASFPSA**DETD**GNLP |
| Opossum | QLRAKLLDLSDSTSVTSFPSA**DETD**GNLP |
| Tasmanian devil | QLRAKLLDLSDSTSVTSFPSA**DETD**GNLP |
| Wallaby | QLRSKLSDLLDSGSVASL-S-**DETD**GNLV |
| platypus | QLRRKILDLLDSTSVASLQ–**DETD**GNLS |
| Birds | |
| Saker falcon | QLRRKILDLLDSTSVASLQ–**DETD**GNLS |
| Peregrine falcon | QLRRKIMDLLDSTSVASLP–**DE**m**D**GNLP |
| Collared flycatcher | QLRRKIMDLLDSTSVASLP–**DETD**GNLS |
| White-throated sparrow | QLRRKIMDLLDSTSVASLP–**DETD**GNLS |
| Medium ground finch | QLRRKIMDLLDSTSVASLP–**DETD**GNLS |
| Zebra finch | QLRRKIMDLLDSTSVASLP–**DETD**GNLS |
| Tibetan ground jay | QLRRKILDLLDSTSVASLQ–**DETD**GNLS |
| Budgerigar | QLRRKILDLLDSTSVASLQ–**DETD**GNLS |
| Parrot | QLRRKILDLLDSTSVASLQ–**DETD**GNLS |
| Scarlet macaw | QLRRKILDLLDSTSVASLQ–**DETD**GNLS |
| Rock pigeon | QLRRKILDLLDSTSVASLP–**DETD**GNLS |
| Mallard duck | QLRRKILDLLDSTSVASLQ–**DETD**GNLS |
| Chicken | QLRRKILDLLDSTSVSSLQ–**DE**i**D**GNLS |
| Turkey | QLRRKILDLLDSTSVSSLQ–**DE**i**D**GNLS |
| *Sarcopterygii* | |
| American alligator | QLRSKILDLLDSTSVASLQ–**DETD**GNVS |
| Green seaturtle | QLRSKILDLLDSTSVASLQ–**DETD**GNLT |
| Painted turtle | QLRSKLLDLLDSTSVASLQ–**DETD**GNLT |
| Chinese softshell turtle | QLRSKILDLLDSASVASLQ–**DETD**GNLT |
| Spiny softshell turtle | QLRSKILDLLDSASVASLQ–**DETD**GNLT |
| Lizard | QLRSKILDLLDSTSVASLP–**DETD**GNIV |
| *X. tropicalis* | QLRARLADLLDSTSVASLQ–**D**d**TD**GIIG |
| Coelacanth | QLRAKLMDLLDNTSVASLQ–**DE**v**D**GNVI |
| Fish | |
| Tetraodon | QLREKLNDLLDNSSITSLQ–**DETD**SNIA |
| Fugu | QLREKLNDLLDNSSITSLQ–**DETD**SNIA |
| Yellowbelly pufferfish | QLREKLNDLLDNSSITSLQ–**DETD**SNIA |

*Table 1 continued on next page*

| Nile tilapia | QLREKLNDLLDNSSVTSLQ–**DETD**SNIA |
| Princess of Burundi | QLREKLNDLLDNSSVTSLQ–**DETD**SNIA |
| Burton's mouthbreeder | QLREKLNDLLDNSSVTSLQ–**DETD**SNIA |
| Zebra mbuna | QLREKLNDLLDNSSVTSLQ–**DETD**SNIA |
| *Pundamilia nyererei* | QLREKLNDLLDNSSVTSLQ–**DETD**SNIA |
| Medaka | QLREKLNDMLENSSITSLQ–**DETD**SNIA |
| Southern platyfish | QLREKLSDLLENSSVTSLQ–**DETD**SNTA |
| Stickleback | QLREKLNDLLDTSSVTSLQ–**DETD**GNIA |
| Atlantic cod | QLREKLNDLLDSSSVTSLL–**DETD**GNLA |
| Zebrafish | QLREKLNDLLDNSSVTSLQ–**DE**|**D**SNIA |
| Spotted gar | QLREKLNDLLDNSSVASLQ–**DE**a**D**GNIA |

Protein sequences orthologous to human ROCK1 were aligned in 93 vertebrate species using Multiz Alignment (**Blanchette et al., 2004**) in the UCSC Genome Browser. These include representative species from the major classes: 62 mammals, 14 birds, 8 *Sarcopterygii* and 14 fish. The conserved caspase cleavage site indicated with underlined bold typeface, with non-identical amino acids in lowercase.

## mESC screening for homologous recombination

Screening of G418-resistant mESCs for homologous recombination was accomplished with separate PCR reactions against the 3′ and 5′ homology arms of the insert vector. Spotted genomic DNA was diluted in 50 µL TE buffer (Tris 10 mM pH 8.0 and EDTA 1 mM) for 24 hr at 37°C and then screened, remaining DNA was re-frozen. For the primary screen tested for recombination in the 3′ targeting vector homology arm and used primers within the neomycin selection cassette and outside the homology arm (*Figure 1—figure supplement 2B*), this reaction was expected to yield a 3.5 kb product in correctly recombined samples. PCR reaction used 200 ng template DNA, one unit Phire hotstart DNA polymerase (Finnzymes), 0.2 mM dNTPs and 1 µM primers. Positive samples were re-analysed. In total, 200 clones were screened, and three samples verified positive for 3′ recombination were then tested for 5′ homology arm recombination and used primers outside the homology arm and within the neomycin selection cassette (*Figure 1—figure supplement 2C*), ultimately confirming two properly recombined clones. This secondary screening reaction was expected to yield a 5.5 kb product in correctly recombined samples. The PCR reaction used the Expand high-fidelity PCR system (Roche) with 400 ng template DNA, one unit polymerase, 0.5 mM dNTPs and 500 nM primers. All PCR reactions were examined for product by agarose gel electrophoresis visualized with ethidium bromide. Product sizes were estimated from DNA ladders (Hyperladder I, Bioline). Successfully targeted clones were then expanded, and stocks generated for subsequent injection into mouse blastocysts.

## mESC blastocyst injection and embryo implantation

Targeted stem cells were grown on MEF monolayers and harvested for injection following plating out of MEFs. Stem cells (8–15) were injected into C57Bl/6 blastocysts and then implanted into the uteri of pseudo-pregnant female ICR mice. Male chimeras of high ESC contribution were identified in the subsequent litters by coat colour. These were crossed with C57Bl/6 females and the offspring genotyped for ROCK1nc. Genotypically positive mice were then crossed to a Cre-deleter strain (on a C57Bl/6 background) to remove the neomycin selection cassette. Mice homozygous for ROCK1nc were then interbred to generate a homozygous strain. This work was performed by the Beatson Institute Transgenic Technology core service.

## Animal genotyping

For routine genotyping, mice were ear notched at weaning and samples sent to Transnetyx genotyping service for analysis.

## MEF generation

MEFs were generated from E13.5 embryos resulting from homozygous ROCK1wt and ROCK1nc breeding pairs. The appearance of the mating plug indicated embryonic day 0.5 (E0.5), and 13 days later pregnant females were euthanized by cervical dislocation and uterine horns containing embryos extracted. Briefly, the embryos were dissected from the uterus and washed in PBS. Embryonic heads and livers were removed and used for genotyping. The remaining tissue from all embryos was pooled and roughly chopped before incubation in 0.05% trypsin (1 mL trypsin/embryo) for 10 min at 37°C. Embryos were then further dissociated by trituration in serological pipettes of decreasing size, then counted and plated at $5 \times 10^6$ cells in 180 cm flasks. Cells were cultured in DMEM supplemented with 10% FBS, penicillin/streptomycin (100 units/mL, 100 µg/mL, respectively) and maintained at 37°C in 5% $CO_2$ atmosphere. Cells were split twice before cryopreservation. Typically, ROCK1wt and ROCK1nc MEF cultures proliferated to passage 8–10 before senescing. No differences in proliferation rate or senescence were observed between ROCK1 genotypes.

## In-cell western blot

MEFs (50,000/well) were plated in 96-well plates and serum-starved overnight before stimulation with 10% FBS for 5 min ±10 µM Y27632. Staining for pMLC was performed as previously described (*Unbekandt et al., 2014*), using rabbit anti-pMLC2 Thr18/Ser19 (#3674; Cell Signaling Technology), and detected with goat anti-rabbit IgG Alexa Fluor 680 (A12076; Invitrogen). Total cell number was determined by nuclei staining with the fluorescent marker DRAQ5 (1:700, Biostatus). Fluorescence was determined by imaging the plate with an Odyssey LiCOR scanner and signal intensity with Odyssey software. pMLC signal was normalized to DRAQ5 signal.

## Western blotting

Apoptotic ROCK1wt and ROCK1nc MEFs were left untreated or treated with TNFα (50 ng/mL) and CHX (10 µg/mL) as indicated. Four hours after treatment, cells were collected by scraping, centrifuged and lysed in RIPA buffer (10 mM Tris pH 7.5, 5 mM EDTA, 150 mM NaCl, 40 mM NaPPi, 50 mM NaF, 1% (v/v) NP-40, 0.5% (v/v) sodium deoxycholate, 0.025% (w/v) SDS, 1 mM $Na_3VO_4$, 1 mM PMSF). Samples were normalized to total protein concentration determined by Bradford colorimetric assay and separated on 10% SDS-PAGE gels. StrataClean resin (Agilent) was used to concentrate protein released by dead and dying cells as we previously described (*Wickman et al., 2013*). Recombinant GST-PAK2 (*Banko et al., 2011*) was expressed in *Escherichia coli* and purified using standard methods (*Cameron et al., 2015*). To control for protein recovery, 2 µg of GST-PAK was added to 2 mL of conditioned media collected from equal numbers of ROCK1wt or ROCK1nc MEFs that were left untreated or treated with TNFα plus CHX for 24 hr. Proteins were recovered from StrataClean resin by boiling for 5 min in sample buffer prior to separation on 10% SDS-PAGE gels. Gels were transferred to nitrocellulose blotting paper and blocked with BSA. Multiple membranes were probed for ROCK1 (BD-Transduction Labs), PARP1 (BD Pharmingen), cleaved PARP1 (Cell Signaling), pMLC (Cell Signaling), cleaved caspase 3 (Abcam), HMGB1 (Cell Signaling) or α-tubulin (Santa Cruz). Antibody binding was visualized and quantified with fluorescent secondary antibodies using an Odyssey LiCOR scanner.

## Caspase activity assay

MEFs were seeded on 96-well plates, then serum-starved overnight prior to induction of apoptosis with TNFα (50 ng/mL) plus CHX (10 µg/mL) or anti-CD95 antibody (2 µg/mL) plus CHX (10 µg/mL) for 4 hr. Cells were subjected to caspase 3/7 activity measurement using a Caspase-Glo assay kit. Briefly, plates containing cells were removed from the incubator and allowed to equilibrate to room temperature for at least 30 min. 100 µL of Caspase-Glo reagent was added to each well, contents mixed gently and then incubated for 30 min at room temperature. The luminescence of each sample was measured in a plate-reading luminometer.

## Flow cytometric determination of phosphatidyl serine (PS) externalization

MEFs were plated on 10 cm plates at a density of $1.5 \times 10^6$ cells/ plate. Cells were serum-starved overnight prior to induction of apoptosis with TNFα plus CHX diluted in serum-free starvation

media. After 4 hr of treatment, cells were collected, centrifuged and diluted to $1 \times 10^5$ cells/mL in a total volume of 100 µL in binding buffer (100 mM HEPES pH 7.4, 140 mM NaCl, 2.5 mM $CaCl_2$) prior to staining with 5 µg/mL propidium iodide (PI) and 5 µL Annexin V-Alexa Fluor 488 conjugate (Abcam) for 10 min at room temperature. Samples were diluted to 0.5 mL in binding buffer, then 10,000 apoptotic cells were analysed by flow cytometry and gated using forward scatter and side scatter, propidium iodide and annexin fluorescence determined in channel FL1 and FL3. Quadrant gating was determined from stained non-apoptotic MEF samples.

## LDH activity measurements

MEFs were serum-starved overnight prior to induction of apoptosis with continued serum starvation, TNFα plus CHX or anti-CD95 antibody plus CHX. LDH activity was measured with Roche cytotoxicity detection kits according to the manufacturer's recommendations. 100 µL of each sample was mixed with an equal volume of dye solution and incubated for 20 min in the dark in a 96-well plate. The reactions were stopped by addition of 50 µL stop solution and absorbance was measured with a TECAN safire2 at 490 nm. Sample activity was normalized to cell densities determined by crystal violet staining of plates grown in parallel.

ROCK1wt or ROCK1nc MEFs were plated at $1.5 \times 10^4$ per well of 96-well plates. Cells were fixed with 4% paraformaldehyde in PBS for 15 min, permeabilized with 0.5% (v/v) Triton-X100 in PBS for 15 min and blocked in 1% (w/v) BSA in PBS for 30 min. Cells were stained with Alexa Fluor 568 Phalloidin (ThermoFisher), DAPI (SigmaAldrich) and Deep Red Whole Cell Stain (ThermoFisher). MEFs were imaged and analysed using the Operetta high content imaging system as described in *Unbekandt et al., 2018*.

## Time-lapse microscopy of MEFs

ROCK1wt or ROCK1nc MEFs were plated on 6-well glass-bottom dishes at a density of $2 \times 10^6$ cells per well. Cells were serum-starved overnight prior to induction of apoptosis with TNFα (50 ng/mL) and CHX (10 µg/mL) diluted in serum-free starvation medium. DIC time-lapse microscopy images were acquired with a 20× DIC objective using Nikon Eclipse TI microscope with a heated stage and 5% $CO_2$ gas line. Immediately after induction of apoptosis the dishes were transferred to the microscope and time-lapse images were taken each minute.

## Scanning electron microscopy

ROCK1wt or ROCK1nc MEFs were seeded on coverslips pre-coated with collagen in 12-well plates. Cells were serum-starved overnight prior to induction of apoptosis with 50 ng/mL of TNFα and 10 µg/mL of CHX diluted in serum-free medium. After 2 hr, cells were fixed with 2.5% (v/v) glutaraldehyde in 0.1 M phosphate buffer for 1 hr at room temperature. After fixation, cells were washed with phosphate buffer (three washes for 10 min), post-fixed in 1% osmium tetroxide/0.1 M sodium cacodylate buffer for 1 hr, then washed for $3 \times 10$ min in distilled water. Dehydration steps were through a graded ethanol series (30 50, 70, 90%) for 10 min each, then 100% ethanol four times for 5 min each. Samples were dried with hexamethyldisilazane (SigmaAldrich) before coating with gold/palladium and were imaged on a JEOL 6400 SEM running at 10 kV. TIFF images were captured using Olympus Scandium software at the Electron Microscopy Facility at the University of Glasgow.

## Elastic micropillars

PDMS micropillar arrays were created, with height 5 µm and spring constant 2.35 nN/µm. Arrays were coated with 10 µL/mL fibronectin to allow cell attachment. Cells were seeded on top of pillars in serum-containing medium and left to spread for 2 hr. For starvation and apoptosis conditions, the medium was changed to serum-free medium and cells on pillars were left overnight at 37°C. For the apoptosis conditions, the next morning, medium was changed to serum-free medium containing 50 ng/mL TNFα and 10 µg/mL CHX. Videos of 10 min duration were captured at 1 frame per 2 s with a Nikon Eclipse Ti-E microscope with a 40× objective lens. Forces exerted by cells were calculated with a custom MATLAB code that tracks the deflection of each pillar centre over time (*Chronopoulos et al., 2016*). The maximum force applied to each individual pillar in contact with the cell was used to generate the output of mean maximum force per cell.

## Mouse experiments

All mouse experiments were approved by the University of Glasgow College of Medicine, Veterinary and Life Sciences Research Ethics Committee, and performed under a project licence granted by the United Kingdom government Home Office, in line with the Animals (Scientific Procedures) Act 1986 and European Union Directive 2010/63/EU in a dedicated barriered facility. ROCK1wt and ROCK1nc were backcrossed for more than 10 generations onto a C57Bl/6J strain. Mice were assigned to treatment groups by researchers who were not blinded with respect to genotypes.

## Haematology

Mice were euthanized by carbon dioxide inhalation, and blood samples were collected via cardiac puncture in potassium-EDTA tubes. Samples were immediately sent to the Clinical Pathology Lab at the University of Glasgow Veterinary School for complete haematology analysis.

## Histology and immunohistochemistry

Hemosiderin deposition in spleens was stained with Perls Prussian blue stain using an equal part solution of ferrocyanide and hydrochloric acid for 25 min at room temperature before washing and counterstaining with eosin. Kidney sections were stained with PAS kits (SigmaAldrich) according to the manufacturer's directions. After rinsing in distilled water, the slides were then transferred to water for 1 min or until blue colour developed to the desired intensity. Slides were rinsed in distilled water and the sections were dehydrated using increasing concentrations of ethanol (one wash for 3 min in 70% ethanol, two washes of 3 min each in 100% ethanol). Slides were then cleared by three washes in xylene for 3 min each, before mounting with DPX mountant (SigmaAldrich).

## DEN, GLZ and CLI095 preparation and administration

DEN was prepared to a final concentration of 10 mg/mL in PBS. For short-term studies assessing DEN-induced hepatic injury, 10-week-old mice were treated with a single dose of DEN at 100 mg/kg body weight by IP injection and culled at 24, 48, 72 and 96 hr after DEN administration. For long-term induction of liver tumours, 14-day-old mice were treated with a 40 µL single dose of DEN (100 mg/mL) by IP injection. After 6 or 9 months, mice were euthanized and blood and livers collected for analysis.

To prepare GLZ (SigmaAldrich), sterile water was used to dissolve GLZ to a final concentration of 10 mg/mL. The solution was filtered using a 70 µm filter prior to administration by IP injection. For acute GLZ and DEN combination treatment, six doses of 200 mg/kg each of GLZ were given along with a single IP dose of DEN (100 mg/kg). Briefly, one dose of GLZ was given the day before DEN treatment and one dose following DEN treatment, then GLZ was injected twice daily for two more days. The following day (72 hr after DEN administration), mice were sacrificed, and blood and livers were removed for analysis. For GLZ and DEN combination treatments for liver tumour induction, mice were treated as indicated in *Figure 7A*.

CLI095 (InvivoGen) was dissolved in DMSO (SigmaAldrich) at a concentration of 3 mg/mL and vortexed until completely solubilized. Final dosages were diluted 1:10 in PBS to 300 µg/mL. For CLI095 and DEN combination treatment, four doses of 3 mg/kg each of CLI095 were given by IP injection along with a single dose of DEN (100 mg/kg). Briefly, one dose of CLI095 was given the day before DEN treatment and one dose following DEN treatment, then CLI095 was injected once daily for two more days. The following day (72 hr after DEN administration), mice were sacrificed, and blood and livers removed for analysis.

## Liver function biochemistry tests

Mice were euthanized by $CO_2$ inhalation and blood samples were collected via cardiac puncture in 1.5 mL microcentrifuge tubes. Blood was allowed to clot at room temperature for at least 30 min and then centrifuged at 1500 × g for 15 min at 4℃. The resulting supernatant (serum) was transferred to a fresh tube for analysis. If sera were not analysed immediately, they were aliquoted and stored at −20℃ until analysis. Liver function tests were performed by the Clinical Pathology Lab at the University of Glasgow Veterinary School, where serum concentrations of ALT and bilirubin were determined using an automated analyser.

## Tissue collection and fixation

Mice were euthanized by $CO_2$ inhalation; weights were recorded and blood collected for haematology or biochemistry analysis. Necropsy was performed immediately to prevent tissue autolysis. Tissues collected were fixed for at least 24 hr by immersing in 10% neutral buffered formalin. For acute liver studies, the left lobe of the liver was routinely used for histological analysis and fixed in 10% neutral buffered formalin. Small pieces from the median lobe were frozen using dry ice.

For tumour studies, mice were examined for visible lesions in all organs. For DEN-induced HCC studies, liver weights were recorded and the number of tumours counted. Tumour foci were measured using digitized Vernier callipers. Tumour volumes were calculated using a modified ellipsoid formula: volume = (length $\times$ width$^2$)/2 (*Euhus et al., 1986*). Lungs were also collected to analyse metastatic lesions. Livers and lungs were fixed in 10% neutral buffered formalin for 24 hr for histological analysis.

## Histology and immunohistochemistry

Paraffin-embedded tissue sections of 4 µm thickness were cut and routinely stained with H&E by the Cancer Research UK Beatson Institute Histology Service. Blank paraffin sections were stored at 4°C until ready for use. IHC was routinely performed on 4-µm-thick formalin-fixed paraffin- embedded (FFPE) sections. IHC for F4/80 and S100A9 was performed by the Histology Service using the following conditions: F4/80 (Abcam), antigen retrieval with Proteinase K at room temperature for 10 min, primary antibody staining for 35 min at room temperature at 1:400 dilution; S100A9 (Santa Cruz), antigen retrieval with citrate buffer pH 6 at 98°C for 20 min, primary antibody staining for 35 min at 1:1000 dilution at room temperature. For secondary antibody staining, either ImmPRESS kits (Vector Labs) or Envision Reagents (Dako) were used by the Histology Service.

HMBG1 staining was performed as follows: sections were deparaffinized in xylene (three washes of 5 min each) and rehydrated through decreasing concentrations of ethanol (two washes of 3 min each in 100% ethanol, one wash for 3 min in 70% ethanol). Slides were then washed in distilled water ($dH_2O$). To unmask antigen, slides were boiled at 98°C in a water bath for 20 min in citrate buffer pH 6. Slides were left to cool down to room temperature for at least 30 min and then rinsed in distilled water. Endogenous peroxidase activity was quenched by incubating sections in 3% $H_2O_2$ in PBS for 10 min. After slides were rinsed once in distilled water and once in TBST, the sections were blocked using blocking solution containing 5% goat serum in Tris-buffered saline/Tween-20 (TBST; 20 mM Tris pH 7.5, 150 mM NaCl, 0.1% (v/v) Tween-20) for 30 min at room temperature. Primary antibody (Rabbit monoclonal anti-HMGB1, Abcam) diluted 1:100 in blocking solution was added to the slides and kept in a humidified chamber overnight. Slides were rinsed three times in TBST to remove excess/unbound primary antibody. HRP-labelled polymer anti-rabbit secondary antibody (Dako) was added to the slides and incubated for 45 min at room temperature. Slides were rinsed again three times in TBST before staining with 3,3'-diaminobenzidine tetrahydrochloride (DAB), as chromogen. Sections were incubated with DAB solution until brown colour developed and the slides immediately transferred to TBST to stop the reaction. For counterstaining, slides were rinsed in distilled water, then stained with haematoxylin for 1 min. In all antibody staining experiments, omitting the primary antibody on one slide served as a negative control.

Collagen deposition in liver sections was detected using Sirius Red and Fast Green as a counterstain as it binds to non-collagenous proteins. Sections were analysed using an Olympus BX51 brightfield microscope and representative images were obtained at different magnifications.

## TUNEL staining

TUNEL staining was performed using in situ cell death detection kits (SigmaAldrich). FFPE tissue sections were deparaffinized and rehydrated, then incubated in Proteinase K solution for 15 min at room temperature. TUNEL reaction mixture was prepared according to the manufacturer's instructions by mixing 50 µL of Enzyme solution and 450 µL of Label solution. The slides were rinsed twice in PBS, and 50–100 µL (depending on the size of the section) of TUNEL reaction mixture was added to the sections. Label solution alone was added to negative control slides. The slides were incubated in a humidified chamber for 60 min at 37°C in the dark. Slides were then rinsed three times in PBS before mounting with Vectashield DAPI (hard set). Sections were imaged using an Olympus BX51 fluorescence microscope using a 20× objective lens.

## Zymosan-induced peritonitis

Zymosan (SigmaAldrich) was dissolved in sterile PBS by vigorous shaking to allow even distribution of zymosan particles at a concentration of 2 mg/mL. Mice were subjected to peritoneal injection with 500 µL zymosan solution (1 mg per mouse). For peritoneal lavage, 5 mL of ice-cold lavage buffer (PBS with 2 mM EDTA) was injected into the peritoneal cavity. Mice abdomen were gently massaged to dislodge cells before collecting the fluid into ice-cold collection tubes, which was then passed through 70 µm filters. The lavage fluid was centrifuged at 1500 rpm for 8 min, then the cell pellet was re-suspended in FACS buffer for haematology analysis by the Clinical Pathology Lab at the University of Glasgow Veterinary School.

## Acknowledgements

This study was funded by the Cancer Research UK institutional funding to the CRUK Beatson Institute (A10419, A17196) and to the Olson lab (A18276). Additional funding to MFO from the Canadian Institutes of Health Research (PJT-169125), Natural Sciences and Engineering Research Council of Canada (RGPIN-2020-05388) and Canada Research Chairs Program (950-231665). Thanks to the Cancer Research UK Beatson Institute Biological and Histology Services, as well as the University of Glasgow Veterinary School Clinical Pathology Laboratory. Thanks to Ayala King (CRUK Beatson Institute) and Robert Nibbs (University of Glasgow) for helpful discussions.

## Additional information

### Funding

| Funder | Grant reference number | Author |
| --- | --- | --- |
| Cancer Research UK | A10419 | Linda Julian<br>Gregory Naylor<br>Grant R Wickman<br>Nicola Rath<br>David Stevenson<br>Sheila Bryson<br>June Munro<br>Lynn McGarry<br>Michael F Olson |
| Cancer Research UK | A17196 | Linda Julian<br>Gregory Naylor<br>Grant R Wickman<br>Nicola Rath<br>David Stevenson<br>Sheila Bryson<br>June Munro<br>Lynn McGarry<br>Michael F Olson |
| Cancer Research UK | A18276 | Linda Julian<br>Gregory Naylor<br>Grant R Wickman<br>Nicola Rath<br>Michael F Olson |
| Canada Research Chairs | 950-231665 | Michael F Olson |
| Natural Sciences and Engineering Research Council of Canada | RGPIN-2020-05388 | Michael F Olson<br>Giovanni Castino |
| Canadian Institutes of Health Research | PJT-169106 | Michael F Olson<br>Giovanni Castino |

The funders had no role in study design, data collection and interpretation, or the decision to submit the work for publication.

## Author contributions
Linda Julian, Gregory Naylor, Grant R Wickman, David Stevenson, Data curation, Formal analysis, Validation, Investigation, Visualization, Methodology, Writing - original draft, Writing - review and editing; Nicola Rath, Giovanni Castino, June Munro, Alistair Rice, Data curation, Formal analysis, Validation, Investigation, Visualization, Methodology; Sheila Bryson, Methodology; Lynn McGarry, Data curation, Formal analysis, Validation, Investigation, Methodology; Margaret Mullin, Data curation, Investigation, Methodology; Armandodel Del Río Hernández, Resources, Supervision, Methodology; Michael F Olson, Conceptualization, Data curation, Formal analysis, Supervision, Funding acquisition, Visualization, Methodology, Writing - original draft, Project administration, Writing - review and editing

## Author ORCIDs
Lynn McGarry http://orcid.org/0000-0002-7055-2615
Michael F Olson https://orcid.org/0000-0003-3428-3507

## Ethics
Animal experimentation: All mouse experiments were approved by the University of Glasgow College of Medicine, Veterinary and Life Sciences Research Ethics Committee, and performed under a project license granted by the United Kingdom government Home Office, in line with the Animals (Scientific Procedures) Act 1986 and European Union Directive 2010/63/EU in a dedicated barriered facility.

## Decision letter and Author response
Decision letter https://doi.org/10.7554/eLife.61983.sa1
Author response https://doi.org/10.7554/eLife.61983.sa2

# Additional files

## Supplementary files
• Transparent reporting form

## Data availability
Data generated during this study are presented in the figures. No datasets were generated or used in this study.

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
