## [Decision Letter]

**Acceptance summary:**

Blebbing of apoptotic cells is a conserved process but its physiological role is not clear. This paper examines the impact of disrupting apoptotic blebbing by introducing a ROCK mutation at Asp 1113 that prevents caspase cleavage-dependent constitutive activation but not normal ROCK1 activity in cell culture and in mice. Surprisingly, disrupting apoptotic blebbing has minimal effects in mice. However, in response to liver damage, there is a pronounced increase in neutrophils and, over the long-term, this translates to fewer hepatocellular carcinomas. This is presumably due to the increased DAMPs that promote clearance of necrotic cells, since markers of apoptosis are not altered.

**Decision letter after peer review:**

Thank you for submitting your article "Defective apoptotic cell contractility provokes sterile inflammation leading to liver damage and tumour suppression" for consideration by *eLife*. Your article has been reviewed by 2 peer reviewers, including Jody Rosenblatt as the Reviewing Editor and Reviewer #3, and the evaluation has been overseen by Carla Rothlin as the Senior Editor.

The reviewers have discussed the reviews with one another and the Reviewing Editor has drafted this decision to help you prepare a revised submission.

Summary:

Blebbing of apoptotic cells is a conserved process but its physiological role is not clear. This paper examines the impact of disrupting apoptotic blobbing by introducing a ROCK mutation at Asp 1113 that prevents caspase cleavage-dependent constitutive activation but not normal ROCK1 activity in cell culture and in mice. Surprisingly, disrupting apoptotic blebbing has minimal effects in mice. However, in response to liver damage, there is a pronounced increase in neutrophils and, over the long-term, this translates to fewer hepatocellular carcinomas. This is presumably due to the increased DAMPs that promote clearance of necrotic cells, since markers of apoptosis are not altered.

Essential revisions:

While the reviewers found this paper interesting for a wide-readership, they felt that the conclusions could be better supported with clearer analysis (see below). One of the main points is that it is not clear how disrupting blebbing may cause secondary necrosis and the measurements for this were not completely convincing, as highlighted by Reviewer #2. The tumor studies also seem to require more statistical power and sufficient explanation, as they are currently counter intuitive and confusing. In the long-term, is there a physiological impact on death from cancer? For greater clarity and because we only have two reviewers, I have left the specific comments below to help you address these points, which we do not believe should create undue workload.

Reviewer #2:

In this paper, the authors have generated a mouse in which ROCK1 is not cleaved during apoptosis, owing to mutation of the caspase-cleavage site (ROCK1nc). They have confirmed that this cleavage is necessary for cell contraction during apoptosis. in vivo, the mice seem to have no phenotype, but do seem to show increased liver damage in response to DEN, with some effect on numbers of tumors. While there are some important weaknesses in the data, as noted below, much of the work is robust and the observations are reported appropriately. I do have issues with the interpretation, also noted below.

1. The data in Figure 1 are convincing and the experiments are well performed and interpreted, with the exception of the interpretation of Figure 1F. The rather small differences in LDH release are readily attributable to different cell lines (it is unclear how many times this was repeated with cells prepared from different animals). The main point, though, is that LDH is released in all cases (as expected at this time point) and there may be a little more when ROCK1 is not cleaved. It is possible that this effect might be more robust at earlier time points, but I don't see why ROCK1 cleavage should have anything to do with preventing secondary necrosis, per se, and LDH release is not necessarily the best indication of this (for example, while AnnV staining is unaffected, is the loss of plasma membrane integrity actually accelerated?). I would say simply tone down the interpretation, except it is at the center of their later conclusions, and this reviewer is not convinced.

2. Figure 2 is overall very nice, and the effects convincing. However, it seems that the difference reported in 2E is predominantly due to two WT cells (two others are in the same range as all the other cells). More numbers are needed here to make this convincing.

3. In Figure 3, the numbers are from "3-11" mice per group in 3A. It would be helpful if the data were presented as individual measurements with mean{plus minus}SD overlayed. This would help us see how many animals are represented at the most relevant time points. This applies to Figure 4 as well. These figures are integral to the paper, as they represents the data that support one of the key conclusions.

4. Figure 5A is meant to imply that the cleavage of ROCK1 prevents cytosolic localization of HMGB1 during apoptosis. This is potentially interesting, but clearly, if this is important, it should be readily demonstrated in vitro, where HMGB1 localization can be much more quantitatively evaluated. Since they wish to conclude that increased release of HMGB1 is responsible for the increased neutrophil infiltration, I have to ask-is more HMGB1 released from the ROCK1nc cells than WT during apoptosis? Again, this is readily evaluated in vitro.

5. While Glycyrrhizin binds to HMGB1 with low affinity (in the range of 0.15 mM) and may inhibit its immunostimulatory effects, this extract of black licorice (also present in many traditional Chinese medicines) is by no means a specific inhibitor of HMGB1. That it has an anti-inflammatory effect is known and is clear in the presented data, but this result is wildly over-interpreted. The effect of TLR4 inhibition by CL1095 is also clear, but there are many explanations for this, in vivo, beyond the conclusion that it is blocking HMGB1 activation of TLR4. Please do not over-interpret this result.

6. Why should GLZ or CL1095 inhibit apoptosis in the ROCK1cn mice treated with DEN? (Figure 5c). I can think of no rationale for this, unless the TUNEL positivity is predominantly from dying neutrophils.

7. Tumor number but not volume appear to be decreased in DEN treated ROCK1nc mice (Figure 6C,D), and this experiment is sufficiently powered to draw this conclusion (at least at 9 months). The authors also suggest that numbers of lung mets at 6 (but not 9) months are reduced (Figure 6E) however it is not clear that this result is sufficiently powered (5 mice/group) to claim a p value of less than 0.0001 when the difference is a mean of 3.5 vs 1.5 (this looks like classic type 2 statistical error). These latter results should not be over-interpreted--there are no evidence that numbers of mets are reduced based on these data.

8. The results in Figure 6 and 7 are interesting, but also confusing. Increased inflammation, necrosis, and neutrophil infiltration would be expected to increase, not decrease HCC. Neutrophils are reportedly required for HCC in the DEN model (e.g. PMID 25879839). The author's suggestion is that increased tumors arise from cells that engage enough active caspase-3 to cleave ROCK1 but somehow survive, and if ROCK1 cannot be cleaved these cells die. I know it is speculation, but it is without basis. The evidence (including that of the authors) is clear that ROCK1 cleavage does not affect the propensity for cell death during apoptosis. Are the authors suggesting that ROCK1 cleavage is necessary for survival? (This relates to the phenomenon of anastasis, which is somewhat controversial, but I guess it is possible). I understand that the authors are only trying to explain their somewhat paradoxical results, but I suggest they should pitch another explanation if possible.

Reviewer #3:

The manuscript, 'Defective apoptotic cell contractility provokes sterile inflammation leading to liver damage and tumour suppression' by Julian et al. reveals interesting physiological roles for apoptotic blebbing in response to liver damage and cancer growth. Blebbing of apoptotic cells is a conserved process but its role is not very clear and they examine the impacts in cell culture and in a mouse model. They show very beautifully that a mutation at Asp 1113 in ROCK1 prevents the caspase cleavage-dependent but not normal ROCK1 activity, resulting in a lack of blebbing. Surprisingly, this mutation in a mouse causes little to no phenotypes, suggesting that its role is subtle.

However, in response to liver damage, there is a pronounced increase in neutrophils and, over the long-term, this translates to fewer hepatocellular carcinomas. This is presumably due to the increased DAMPs that promote clearance of necrotic cells, since markers of apoptosis are not altered. I think that this is an interesting story and well-written. While I enjoyed the historical perspective, overall, I felt that there was a bit too much data and some of the results that showed no difference could be shifted to supplement to tighten up the story.

I have only a few other questions:

1. In figure 1, although the graphs show that there is no significance from the mutant, there does seem to be a surprising trend of having higher, Annexin V, capase and LDH in the NC mutant. Could there be some reason for this. Do they accumulate more over time?

2. In Figure 3 B, the H&E, the mutant seems to have higher cellularity. I can't really tell because it is so low mag but is there some other difference in the structures outside of the Steatosis?

3. Why is there a different neutrophil response between zymosan and DEN? I am not sure what the point of this is in the paper.

4. In Figure 6 A, are the livers in the WT always smaller but more tumour filled?

5. Figure 7B could be moved into supplement, as there are no differences, however, I would like to see examples of the tissues for each score.

6. At the end of the day, does it matter that there are fewer tumours in the liver? Does it change the Kaplan-Meier curve? If not, I might limit what you say about its relevance to cancer. You might expect that it could help launch more of an immune response but since over time they develop the same number of metastases, that doesn't seem to be the case.

[Editors' note: further revisions were suggested prior to acceptance, as described below.]

Thank you for submitting your article "Defective apoptotic cell contractility provokes sterile inflammation leading to liver damage and tumour suppression" for consideration by *eLife*. Your article has been reviewed by 1 peer reviewers, and the evaluation has been overseen by a Reviewing Editor and Carla Rothlin as the Senior Editor. The reviewers have opted to remain anonymous.

We have discussed with one expert in the field your response to the previous reviews and find that there are central points that remain to be addressed. For clarity, we have included their previous question, your response and their recent response, below. Please note that experimental evidence is essential to support the main conclusions of the manuscript.

As the editors have judged that your manuscript is of interest, but as described below that essential experiments are required before it can be considered for publication, we would like to draw your attention to changes in our revision policy that we have made in response to COVID-19 (https://elifesciences.org/articles/57162). First, because many researchers have temporarily lost access to the labs, we will give authors as much time as they need to submit revised manuscripts. We are also offering, if you choose, to post the manuscript to bioRxiv (if it is not already there) along with this decision letter and a formal designation that the manuscript is "in revision at *eLife*". Please let us know if you would like to pursue this option. (If your work is more suitable for medRxiv, you will need to post the preprint yourself, as the mechanisms for us to do so are still in development.)

Summary:

Blebbing of apoptotic cells is a conserved process but its physiological role is not clear. This paper examines the impact of disrupting apoptotic blebbing by introducing a ROCK mutation at Asp 1113 that prevents caspase cleavage-dependent constitutive activation but not normal ROCK1 activity in cell culture and in mice. Surprisingly, disrupting apoptotic blebbing has minimal effects in mice. However, in response to liver damage, there is a pronounced increase in neutrophils and, over the long-term, this translates to fewer hepatocellular carcinomas. This is presumably due to the increased DAMPs that promote clearance of necrotic cells, since markers of apoptosis are not altered.

Reviewer #2:

My review is based on the previous review and the authors' responses.

Previous comment: Figure 5A is meant to imply that the cleavage of ROCK1 prevents cytosolic localization of HMGB1 during apoptosis. This is potentially interesting, but clearly, if this is important, it should be readily demonstrated in vitro, where HMGB1 localization can be much more quantitatively evaluated. Since they wish to conclude that increased release of HMGB1 is responsible for the increased neutrophil infiltration, I have to ask-is more HMGB1 released from the ROCK1nc cells than WT during apoptosis? Again, this is readily evaluated in vitro.

Response: This suggested experiment is very good, and we did attempt to address this question. Unfortunately, the isolation of primary hepatocytes from ROCK1nc and ROCK1wt mice did not yield sufficient numbers of viable cells to perform this biochemical experiment.

New comment: I do not see why this experiment would require the use of primary hepatocytes (although nice, even showing this in MEF would be helpful). The authors conclude that the failure to cleave ROCK1 leads to more release of HMGB1 from cells during apoptosis, and provide no evidence that this is the case. As this is a major conclusion of the paper, it requires more evidence to support it.

Previous comment: The results in Figure 6 and 7 are interesting, but also confusing. Increased inflammation, necrosis, and neutrophil infiltration would be expected to increase, not decrease HCC. Neutrophils are reportedly required for HCC in the DEN model (e.g. PMID 25879839). The author's suggestion is that increased tumors arise from cells that engage enough active caspase-3 to cleave ROCK1 but somehow survive, and if ROCK1 cannot be cleaved these cells die. I know it is speculation, but it is without basis. The evidence (including that of the authors) is clear that ROCK1 cleavage does not affect the propensity for cell death during apoptosis. Are the authors suggesting that ROCK1 cleavage is necessary for survival? (This relates to the phenomenon of anastasis, which is somewhat controversial, but I guess it is possible). I understand that the authors are only trying to explain their somewhat paradoxical results, but I suggest they should pitch another explanation.

Response: In order to clarify the hypothesis that was being tested, the manuscript has been amended on page 15 lines 332-341. "Given that acute DEN treatment resulted in greater liver damage and hepatocyte cell death in ROCK1nc mice relative to ROCK1wt mice (Figure 4), a possibility was that more mutated hepatocytes that might have given rise to liver tumours were killed by DEN in ROCK1nc livers than in ROCK1wt livers. To test this hypothesis. ROCK1nc mice were treated with GLZ or DEN alone, or with the combination as shown in Figure 8A over the 4 day tumour initiation period, which recapitulates the GLZ dosing regimen that reduced neutrophil recruitment (Figure 5B), hepatocyte death (Figure 5C) and liver damage as determined by serum alanine transaminase levels (Figure 5D). After 9 months, during which there was no further treatment with either DEN or GLZ, the number of liver tumours induced by DEN plus GLZ was significantly higher than DEN alone, with GLZ treatment alone having no effect on tumour numbers (Figure 8B)."

New comment: The issue I continue to have is with this statement: "a possibility was that more mutated hepatocytes that might have given rise to liver tumours were killed by DEN in ROCK1nc livers than in ROCK1wt livers. To test this hypothesis, ROCK1nc mice were treated with GLZ or DEN alone, or with the combination." This does not test the hypothesis. The data show that there is less death upon treatment with GLZ or the TLR4i (Figure 5b), and therefore it is not the DEN that kills the hepatocytes but apparently the neutrophils (unless, as I mentioned, they are actually scoring apoptotic neutrophils, which I cannot assess from these images). As I continue to maintain, there is no good reason to think that a failure to cleave ROCK1 would prevent apoptosis to result in survival of the cells, and no data are presented to even suggest this possibility. I would ask that the authors consider an alternative hypothesis, or simply leave this as a paradox (since, as noted, neutrophil recruitment might be expected to exacerbate, not reduce tumor numbers in this model). As currently stated, I think that readers will be confused by the reasoning here.

---

## [Author Response]

Reviewer #2:In this paper, the authors have generated a mouse in which ROCK1 is not cleaved during apoptosis, owing to mutation of the caspase-cleavage site (ROCK1nc). They have confirmed that this cleavage is necessary for cell contraction during apoptosis. in vivo, the mice seem to have no phenotype, but do seem to show increased liver damage in response to DEN, with some effect on numbers of tumors. While there are some important weaknesses in the data, as noted below, much of the work is robust and the observations are reported appropriately. I do have issues with the interpretation, also noted below.1. The data in Figure 1 are convincing and the experiments are well performed and interpreted, with the exception of the interpretation of Figure 1F. The rather small differences in LDH release are readily attributable to different cell lines (it is unclear how many times this was repeated with cells prepared from different animals). The main point, though, is that LDH is released in all cases (as expected at this time point) and there may be a little more when ROCK1 is not cleaved. It is possible that this effect might be more robust at earlier time points, but I don't see why ROCK1 cleavage should have anything to do with preventing secondary necrosis, per se, and LDH release is not necessarily the best indication of this (for example, while AnnV staining is unaffected, is the loss of plasma membrane integrity actually accelerated?). I would say simply tone down the interpretation, except it is at the center of their later conclusions, and this reviewer is not convinced.

We have now included data from 8 hr post-induction of apoptosis for comparison with the 24 hr data originally included (Figure 1F). What is apparent is that there are no significant differences between ROCK1wt and ROCK1nc cells in each condition at 8 hr, which contrasts with the significant differences observed at 24 hr. This does suggest that the loss of membrane integrity is somewhat accelerated given the differences between the 8 to 24 hour time points, but we would prefer to leave the interpretation on manuscript page 7 lines 159-161 “that the inability of caspases to activate ROCK1 and to induce MLC phosphorylation result in morphological differences favouring secondary necrosis during late apoptosis.”

2. Figure 2 is overall very nice, and the effects convincing. However, it seems that the difference reported in 2E is predominantly due to two WT cells (two others are in the same range as all the other cells). More numbers are needed here to make this convincing.

To clarify these findings, the box and whiskers plots have been re-configured as bar charts indicating means ± SD. It is now clearer that greater than 50% of the apoptotic ROCK1wt cells produced more contractile force than all but one of the apoptotic ROCK1nc cells. It has also been indicated that the force generated by apoptotic ROCK1nc cells was not significantly different from non-apoptotic ROCK1nc cells, in marked contrast to apoptotic ROCK1wt cells that produced more contractile force than any other experimental condition.

3. In Figure 3, the numbers are from "3-11" mice per group in 3A. It would be helpful if the data were presented as individual measurements with mean{plus minus}SD overlayed. This would help us see how many animals are represented at the most relevant time points. This applies to Figure 4 as well. These figures are integral to the paper, as they represents the data that support one of the key conclusions.

All panels in Figures 3 and 4 have been re-drawn to indicate the individual measurements.

4. Figure 5A is meant to imply that the cleavage of ROCK1 prevents cytosolic localization of HMGB1 during apoptosis. This is potentially interesting, but clearly, if this is important, it should be readily demonstrated in vitro, where HMGB1 localization can be much more quantitatively evaluated. Since they wish to conclude that increased release of HMGB1 is responsible for the increased neutrophil infiltration, I have to ask-is more HMGB1 released from the ROCK1nc cells than WT during apoptosis? Again, this is readily evaluated in vitro.

This suggested experiment is very good, and we did attempt to address this question. Unfortunately, the isolation of primary hepatocytes from ROCK1nc and ROCK1wt mice did not yield sufficient numbers of viable cells to perform this biochemical experiment.

5. While Glycyrrhizin binds to HMGB1 with low affinity (in the range of 0.15 mM) and may inhibit its immunostimulatory effects, this extract of black licorice (also present in many traditional Chinese medicines) is by no means a specific inhibitor of HMGB1. That it has an anti-inflammatory effect is known and is clear in the presented data, but this result is wildly over-interpreted. The effect of TLR4 inhibition by CL1095 is also clear, but there are many explanations for this, in vivo, beyond the conclusion that it is blocking HMGB1 activation of TLR4. Please do not over-interpret this result.

We have qualified the conclusions of these experiments from line 299 as follows “These results are consistent with the interpretation that the ROCK1nc mutation shifts hepatocyte cell death towards a more necrotic-like form that results in greater HMGB1-TLR4 dependent neutrophil recruitment and consequent amplification of DEN induced liver damage.”

6. Why should GLZ or CL1095 inhibit apoptosis in the ROCK1cn mice treated with DEN? (Figure 5c). I can think of no rationale for this, unless the TUNEL positivity is predominantly from dying neutrophils.

Neutrophil-mediated injury amplification has been reported in numerous publications, including [Huebener et al. The HMGB1/RAGE axis triggers neutrophil-mediated injury amplification following necrosis. J. Clin. Invest. 125, 539–550 (2015).] The interpretation that we presented on manuscript pages 13-14 is that the significantly reduced neutrophil recruitment in mice treated with GLZ or CLI095 in addition to DEN (Figure 5B) resulted in less liver damage, as indicated by fewer apoptotic cells (Figure 5C) and reduced serum alanine transaminase release (Figure 5D).

7. Tumor number but not volume appear to be decreased in DEN treated ROCK1nc mice (Figure 6C,D), and this experiment is sufficiently powered to draw this conclusion (at least at 9 months). The authors also suggest that numbers of lung mets at 6 (but not 9) months are reduced (Figure 6E) however it is not clear that this result is sufficiently powered (5 mice/group) to claim a p value of less than 0.0001 when the difference is a mean of 3.5 vs 1.5 (this looks like classic type 2 statistical error). These latter results should not be over-interpreted – there are no evidence that numbers of mets are reduced based on these data.

The observations that tumour volumes and number of metastases were not significantly different between ROCK1wt and ROCK1nc mice at both 6 and 9 month time points was indicated on manuscript page 14 lines 312-316. The conclusion to this section on manuscript page 15 lines 329-331 was amended as follows “These results indicate that the principal difference between the liver tumours in ROCK1wt and ROCK1nc mice was related to their initiation rather than growth or progression.”

8. The results in Figure 6 and 7 are interesting, but also confusing. Increased inflammation, necrosis, and neutrophil infiltration would be expected to increase, not decrease HCC. Neutrophils are reportedly required for HCC in the DEN model (e.g. PMID 25879839). The author's suggestion is that increased tumors arise from cells that engage enough active caspase-3 to cleave ROCK1 but somehow survive, and if ROCK1 cannot be cleaved these cells die. I know it is speculation, but it is without basis. The evidence (including that of the authors) is clear that ROCK1 cleavage does not affect the propensity for cell death during apoptosis. Are the authors suggesting that ROCK1 cleavage is necessary for survival? (This relates to the phenomenon of anastasis, which is somewhat controversial, but I guess it is possible). I understand that the authors are only trying to explain their somewhat paradoxical results, but I suggest they should pitch another explanation if possible.

In order to clarify the hypothesis that was being tested, the manuscript has been amended on page 15 lines 332-341. “Given that acute DEN treatment resulted in greater liver damage and hepatocyte cell death in ROCK1nc mice relative to ROCK1wt mice (Figure 4), a possibility was that more mutated hepatocytes that might have given rise to liver tumours were killed by DEN in ROCK1nc livers than in ROCK1wt livers. […] After 9 months, during which there was no further treatment with either DEN or GLZ, the number of liver tumours induced by DEN plus GLZ was significantly higher than DEN alone, with GLZ treatment alone having no effect on tumour numbers (Figure 8B).”

Reviewer #3:The manuscript, 'Defective apoptotic cell contractility provokes sterile inflammation leading to liver damage and tumour suppression' by Julian et al. reveals interesting physiological roles for apoptotic blebbing in response to liver damage and cancer growth. Blebbing of apoptotic cells is a conserved process but its role is not very clear and they examine the impacts in cell culture and in a mouse model. They show very beautifully that a mutation at Asp 1113 in ROCK1 prevents the caspase cleavage-dependent but not normal ROCK1 activity, resulting in a lack of blebbing. Surprisingly, this mutation in a mouse causes little to no phenotypes, suggesting that its role is subtle. However, in response to liver damage, there is a pronounced increase in neutrophils and, over the long-term, this translates to fewer hepatocellular carcinomas. This is presumably due to the increased DAMPs that promote clearance of necrotic cells, since markers of apoptosis are not altered. I think that this is an interesting story and well-written. While I enjoyed the historical perspective, overall, I felt that there was a bit too much data and some of the results that showed no difference could be shifted to supplement to tighten up the story.I have only a few other questions:1. In figure 1, although the graphs show that there is no significance from the mutant, there does seem to be a surprising trend of having higher, Annexin V, capase and LDH in the NC mutant. Could there be some reason for this. Do they accumulate more over time?

It is possible that the apparently greater, albeit not significant, Annexin V staining in ROCK1nc MEFs in Figure 1D is related to the reduced contractile force generation shown in Figure 2, but that would be speculative. In addition, we previously showed that pharmacological inhibition of ROCK activity did not affect Annexin V staining in NIH 3T3 mouse fibroblasts [Coleman, M. L. et al. Membrane blebbing during apoptosis results from caspase-mediated activation of ROCK I. Nat. Cell Biol. 3, 339–345 (2001)]. As a result, this is more likely to be biological variability. Caspase activation was slightly higher in response to anti-CD95, but lower in response to TNFα in ROCK1nc MEFs, but neither difference in activity was significantly different, again suggesting that this is biological variability. On the other hand, LDH release was significantly higher in response to nutrient starvation, TNFα or anti-CD95 after 24 hours for ROCK1nc MEFs relative to ROCK1wt MEFs. This observation is likely biologically relevant, and have included the interpretation on manuscript page 7 lines 159-161 that “that the inability of caspases to activate ROCK1 and to induce MLC phosphorylation result in morphological differences favouring secondary necrosis during late apoptosis.”

2. In Figure 3 B, the H&E, the mutant seems to have higher cellularity. I can't really tell because it is so low mag but is there some other difference in the structures outside of the Steatosis?

On manuscript pages 11-12 lines 244 to 256, the histopathological features were described as follows: “At 24 hours post DEN treatment, H&E stained ROCK1wt and ROCK1nc liver sections exhibited comparable hepatocyte ballooning and mild microvesicular steatosis around the centrilobular regions. […] However, ROCK1nc livers continued to have significantly larger necrotic areas relative to ROCK1wt livers (Figure 3B).”

3. Why is there a different neutrophil response between zymosan and DEN? I am not sure what the point of this is in the paper.

The point of comparing DEN-induced neutrophil recruitment to livers with zymosan-induced recruitment to the peritoneal compartment was to determine whether the observed difference between DEN-treated ROCK1wt and ROCK1nc livers in Figure 4B was likely due to recruitment versus the responsiveness of the neutrophils. Given that zymosan induced the same neutrophil responses in Figure 4D, this indicates that the neutrophils (as well as whole blood cells, monocytes and lymphocytes) respond to the zymosan stimulus comparably in each genotype. The interpretation is that the differences in Figure 4B are due to the way that the hepatocytes die in ROCK1nc livers and release signals that recruit neutrophils. This interpretation is consistent with the differences in HMGB1 translocation shown in Figure 5A. On manuscript page 13 lines 279-282, the conclusion we included was “These results are consistent with the greater neutrophil numbers in ROCK1nc livers being a result of differences in the level of recruitment signal(s), rather than differing responses of neutrophils.”

4. In Figure 6 A, are the livers in the WT always smaller but more tumour filled?

The livers in Figure 6A are representative examples 9 months after tumour initiation with DEN. The liver mass and liver/body mass ratios in Figures 7A and 7C are also for samples collected 9 months after DEN treatment. These data reveal that there are no significant differences between the ROCK1wt and ROCK1nc livers, thus the apparent differences in the photos are merely a reflection of sample to sample variability within the genotypes.

5. Figure 7B could be moved into supplement, as there are no differences, however, I would like to see examples of the tissues for each score.

Assuming that it’s Figure 7D that is being referenced, examples of the histopathological features of DEN-induced HCC that were used to score tumours have now been included in a new Supplementary file 7.

6. At the end of the day, does it matter that there are fewer tumours in the liver? Does it change the Kaplan-Meier curve? If not, I might limit what you say about its relevance to cancer. You might expect that it could help launch more of an immune response but since over time they develop the same number of metastases, that doesn't seem to be the case.

At the commencement of the tumour studies, a decision was made to use fixed endpoints (6 and 9 months) to evaluate the influence of the ROCK1nc mutation, based on numerous previous publications that have utilized DEN and other chemically induced models (reviewed in: Carlessi et al. Chapter 4. Mouse Models of Hepatocellular Carcinoma. In: Tirnitz-Parker, editor. Hepatocellular Carcinoma. Codon Publications; 2019 doi: 10.15586/hepatocellularcarcinoma.2019.ch4). Fixed endpoints do not allow for evaluation of the effect of the experimental conditions on survival, but do reveal other important cancer-relevant aspects, including potential influence on tumour initiation, growth and progression.

[Editors' note: further revisions were suggested prior to acceptance, as described below.]

Reviewer #2:My review is based on the previous review and the authors' responses.Previous comment: Figure 5A is meant to imply that the cleavage of ROCK1 prevents cytosolic localization of HMGB1 during apoptosis. This is potentially interesting, but clearly, if this is important, it should be readily demonstrated in vitro, where HMGB1 localization can be much more quantitatively evaluated. Since they wish to conclude that increased release of HMGB1 is responsible for the increased neutrophil infiltration, I have to ask-is more HMGB1 released from the ROCK1nc cells than WT during apoptosis? Again, this is readily evaluated in vitro.Response: This suggested experiment is very good, and we did attempt to address this question. Unfortunately, the isolation of primary hepatocytes from ROCK1nc and ROCK1wt mice did not yield sufficient numbers of viable cells to perform this biochemical experiment.New comment: I do not see why this experiment would require the use of primary hepatocytes (although nice, even showing this in MEF would be helpful). The authors conclude that the failure to cleave ROCK1 leads to more release of HMGB1 from cells during apoptosis, and provide no evidence that this is the case. As this is a major conclusion of the paper, it requires more evidence to support it.

As suggested by the reviewer, we examined HMGB1 release from ROCK1wt and ROCK1nc MEFs. Data shown in a new Figure 1G (and accompanying Figure 1—figure supplement 1D) reveals that cleavage of the well-validated caspase substrate PARP1 is equivalent in cell lysates 4 hours after treatment with TNFα + cycloheximide in ROCK1wt and ROCK1nc MEFs, and yet there is markedly more HMGB1 released into conditioned media 24 hours after the induction of apoptosis. These results are consistent with Figure 1F that shows significantly greater release of lactate dehydrogenase (LDH) in ROCK1nc MEFs 24 hours after the induction of apoptosis. The manuscript has been revised on lines 162-170 as follows:

“Furthermore, four hours after treatment of ROCK1wt and ROCK1nc MEFs with TNFα plus CHX resulted in comparable cleavage of the well-characterized caspase substate PARP1 (Figure 1G and Figure 1—figure supplement 1D). […] In contrast, recovery of the DAMP high mobility group protein 1 (HMGB1) was markedly greater from apoptotic ROCK1nc MEFs relative to ROCK1wt MEFs Figure 1G and Figure 1-figure supplement 1D).”

Previous comment: The results in Figure 6 and 7 are interesting, but also confusing. Increased inflammation, necrosis, and neutrophil infiltration would be expected to increase, not decrease HCC. Neutrophils are reportedly required for HCC in the DEN model (e.g. PMID 25879839). The author's suggestion is that increased tumors arise from cells that engage enough active caspase-3 to cleave ROCK1 but somehow survive, and if ROCK1 cannot be cleaved these cells die. I know it is speculation, but it is without basis. The evidence (including that of the authors) is clear that ROCK1 cleavage does not affect the propensity for cell death during apoptosis. Are the authors suggesting that ROCK1 cleavage is necessary for survival? (This relates to the phenomenon of anastasis, which is somewhat controversial, but I guess it is possible). I understand that the authors are only trying to explain their somewhat paradoxical results, but I suggest they should pitch another explanation.Response: In order to clarify the hypothesis that was being tested, the manuscript has been amended on page 15 lines 332-341. "Given that acute DEN treatment resulted in greater liver damage and hepatocyte cell death in ROCK1nc mice relative to ROCK1wt mice (Figure 4), a possibility was that more mutated hepatocytes that might have given rise to liver tumours were killed by DEN in ROCK1nc livers than in ROCK1wt livers. To test this hypothesis. ROCK1nc mice were treated with GLZ or DEN alone, or with the combination as shown in Figure 8A over the 4 day tumour initiation period, which recapitulates the GLZ dosing regimen that reduced neutrophil recruitment (Figure 5B), hepatocyte death (Figure 5C) and liver damage as determined by serum alanine transaminase levels (Figure 5D). After 9 months, during which there was no further treatment with either DEN or GLZ, the number of liver tumours induced by DEN plus GLZ was significantly higher than DEN alone, with GLZ treatment alone having no effect on tumour numbers (Figure 8B)."New comment: The issue I continue to have is with this statement: "a possibility was that more mutated hepatocytes that might have given rise to liver tumours were killed by DEN in ROCK1nc livers than in ROCK1wt livers. To test this hypothesis, ROCK1nc mice were treated with GLZ or DEN alone, or with the combination." This does not test the hypothesis. The data show that there is less death upon treatment with GLZ or the TLR4i (Figure 5b), and therefore it is not the DEN that kills the hepatocytes but apparently the neutrophils (unless, as I mentioned, they are actually scoring apoptotic neutrophils, which I cannot assess from these images). As I continue to maintain, there is no good reason to think that a failure to cleave ROCK1 would prevent apoptosis to result in survival of the cells, and no data are presented to even suggest this possibility. I would ask that the authors consider an alternative hypothesis, or simply leave this as a paradox (since, as noted, neutrophil recruitment might be expected to exacerbate, not reduce tumor numbers in this model). As currently stated, I think that readers will be confused by the reasoning here.

The sentences on lines 344-348 that the reviewer has highlighted was not written as clearly as they should have been. The reviewer is correct in pointing out that all the

data collected indicates that ROCK1 cleavage does not make cells intrinsically more or less sensitive to apoptotic stimuli. Our suggestion is that the amplification of the DEN-induced damage by neutrophils in the acute 3-4 day period following administration has a longer term consequence on the number of tumours formed. Specifically, the greater numbers of neutrophils recruited in ROCK1nc livers results in more cell death because of damage amplification that is greater than the direct effect of DEN. The fact that there is more death results in there being fewer tumours 9 months later. GLZ addition reduced the acute neutrophil recruitment and tissue damage, as indicated by the reductions in S100A9 stained cells, TUNEL positive cells and serum alanine transaminase levels (Figure 5B-D). Looking 9 months later, that tissue protective effect of GLZ allows for more tumours to develop from cells that survived and which were mutated by DEN. It isn’t expected that neutrophils would persist over time, the data in Figure 4B shows that the numbers normalize 4 days after DEN treatment. Since DEN administration is done as a single dose, there wouldn’t be a chance for neutrophils to affect tumour development beyond the first few days following drug treatment.

The sentences on lines 344-347 have been amended to: “Given that acute DEN treatment resulted in greater liver damage and hepatocyte cell death in ROCK1nc mice relative to ROCK1wt mice (Figure 4), a possibility was that more mutated hepatocytes, which might have given rise to liver tumours, were killed following DEN administration in ROCK1nc livers than in ROCK1wt livers.”

In addition, the following has been added to lines 358-366: “Taken together, these results indicate that the acute injury amplification associated with higher levels of neutrophil recruitment observed in DEN treated ROCK1nc mice had a long-term benefit in reducing HCC tumour numbers (Figure 8D). […] Thus, the benefit from the ability of ROCK1 activation and consequent cell contraction to limit sterile inflammation and damage amplification during acute xenobiotic-induced tissue-scale cell death comes at the long-term cost of less efficient tumour suppression due to the survival of potential tumour initiating cells.”

A new Figure 8D schematic diagram has been added to illustrate the interpretation of the results.